# AGENTS' ROOM: NARRATIVE GENERATION THROUGH MULTI-STEP COLLABORATION

**Fantine Huot, Reinald Kim Amplayo, Jennimaria Palomaki, Alice Shoshana Jakobovits, Elizabeth Clark & Mirella Lapata**
Google DeepMind
{fantinehuot,reinald,jpalomaki,jakobovits,eaclark,lapata}@google.com

## ABSTRACT

Writing compelling fiction is a multifaceted process combining elements such as crafting a plot, developing interesting characters, and using evocative language. While large language models (LLMs) show promise for story writing, they currently rely heavily on intricate prompting, which limits their use. We propose AGENTS' ROOM, a generation framework inspired by narrative theory, that decomposes narrative writing into subtasks tackled by specialized agents. To illustrate our method, we introduce TELL ME A STORY[1], a high-quality dataset of complex writing prompts and human-written stories, and a novel evaluation framework designed specifically for assessing long narratives. We show that AGENTS' ROOM generates stories that are preferred by expert evaluators over those produced by baseline systems by leveraging collaboration and specialization to decompose the complex story writing task into tractable components. We provide extensive analysis with automated and human-based metrics of the generated output.

## 1 INTRODUCTION

Creating long-form content requires meticulous research, advanced planning, an engaging writing style, and the ability to craft stories that captivate. J.K. Rowling is claimed to have had most of the Harry Potter story planned out before she started writing. She knew there would be seven books, which characters would be important and how they would develop, and which key plot twists would serve the overall story. In addition, she carried out substantial research to create the fictional universe which provides the backdrop of the story. Breaking down a story into distinct sections is typical for longer narratives, with most stories boiling down to a few shared elements like exposition, rising action, climax, falling action, and resolution (Freytag, 1896; Pavis, 1998). Practical guides to writing successful screenplays (Cutting, 2016; Hauge, 2017) outline a similar structure, following the setup, the new situation, progress, complications and higher stakes, the final push, and the aftermath.

Large language models (LLMs) have demonstrated impressive writing capabilities (Yang et al., 2022; Nur Fitria, 2023; Shao et al., 2024; Bai et al., 2024), however, generating long-form content is still a challenge. Well-known problems include maintaining a consistent narrative, tone, or factual accuracy over extended stretches of text (Chakrabarty et al., 2024b; Wang et al., 2023b; Al-abdulkarim et al., 2021; Balepur et al., 2023; Yamshchikov & Tikhonov, 2023), and showcasing a unique voice or humor that makes writing truly memorable. Despite displaying flashes of creativity, they often replicate patterns found in their training data, which hinders the generation of original concepts, plotlines, or phrasing. Added problems include the lack of datasets or benchmarks for long-form writing (Bai et al., 2024) and standardized evaluation criteria for assessing creative writing either by humans or machines (Chhun et al., 2022; 2024; Chakrabarty et al., 2024a).

Existing methods often rely on detailed prompts to guide the generation process (Yang et al., 2022; Xie et al., 2023), prompt chaining (Mirowski et al., 2023; Yang et al., 2022), and planning strategies (Yang et al., 2023; Lee et al., 2024) as a means of breaking down the complex writing task into more manageable components. In this paper, we conceptualize long-form writing as a multi-agent

---

[1]We release the dataset and metrics at: https://github.com/google-deepmind/tell_me_a_story

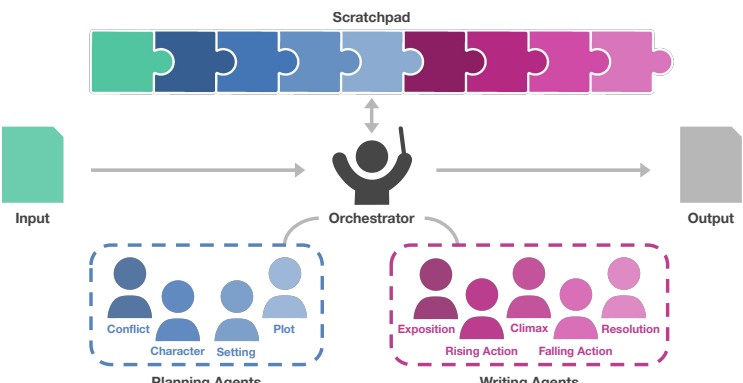

Figure 1: AGENTS' ROOM, a multi-agent framework for collaborative writing. A central orchestrator calls the individual agents and consolidates their contributions into the scratchpad. We color-code each piece of the scratchpad with the contributing agent's color.

collaboration problem. Rather than attempting a decomposition of the writing task within a *single* agent (Chen et al., 2023; Yao et al., 2024), we leverage collaboration among *multiple* agents, with specialized abilities (Talebirad & Nadiri, 2023; Zhang et al., 2024b; Han et al., 2024). We propose AGENTS' ROOM[2] (Figure 1), a generation paradigm which consists of two types of agents, namely *planning* and *writing* agents. Planning agents flesh out key components of the content but do not write the story as such. For example, a planning agent might specialize in character descriptions, whereas another might focus on the plot or central conflict. Writing agents are responsible for generating the final output text and are also specialized, e.g., one may focus on the introduction, and another on the conclusions. The two types of agents work collaboratively to complete the writing task, sharing and managing information through a *scratchpad* which maintains outputs from planning agents and makes them available to writing agents. An *orchestrator* is responsible for calling the agents in order depending on the task at hand.

Compared to single LLM-powered agents, this multi-agent approach offers several advantages:

- LLMs can be specialized into various distinct agents (e.g., zero-shot prompted or fine-tuned) performing a single function with high precision;
- it avoids well-known problems with lengthy and under-specified instructions which require multiple iterations to build context and fully define an appropriate solution;
- it can be applied to problems whose solution is not known beforehand, and results from exploring a vast research space or involves very long output (e.g., writing a book);
- it naturally lends itself to human-in-the loop automation where machine-based agents can be replaced with human ones when needed.

We formalize AGENTS' ROOM as a general writing framework and apply it to creative writing. Specifically, we focus on writing long-form stories (1,000-2,000 tokens), and create specialized agents drawing inspiration from narrative theory (e.g., Card 1999; Noble 1999; Pavis 1998). To evaluate our method, we introduce TELL ME A STORY, a new dataset of human-created writing prompts and fiction stories, and a novel evaluation framework designed for assessing multiple dimensions of story quality. Experimental results show that AGENTS' ROOM generates stories that are preferred (by humans and automatic metrics) over those produced by baseline systems which do not leverage collaboration or specialization.

## 2    RELATED WORK

**Story Generation** The advent of large pre-trained language models has provided a common framework for generating stories which sound fluent but often struggle with maintaining coherence and

---

[2]AGENTS' ROOM is very loosely modeled after writers' room, a collaborative space where writers, (usually of a television series), come together to write and refine scripts.

plausibility. Attempts to enhance coherence and control the trajectory of events often decompose the generation task into planning an outline or sketch, and then elaborating on it, e.g., by filling in descriptions and specific details of each story. Examples of intermediate plans include sequences of entities and their actions (Yao et al., 2019), outlines (Fan et al., 2019; Zhou et al., 2023; Wang et al., 2023a), plot structures (Goldfarb-Tarrant et al., 2020), and more elaborate descriptions including details about the setting of the story, its characters, and main plot points (Yang et al., 2022; 2023). Other work uses common sense knowledge to impose constraints on the characters and their interactions (Peng et al., 2022), ensemble-based models to render event sequences more plausible (Ammanabrolu et al., 2020), stylistic constraints (Kong et al., 2021), and twists through constrained decoding (Huang et al., 2023). These efforts have demonstrated that generating stories as a one-step process is challenging, and ultimately various interventions are required to improve overall story quality. Our work follows on from this realization, and breaks down the writing task into subtasks, undertaken by different agents who collaboratively plan and write a story. Collaborative writing is often used in academic or professional contexts to leverage the strengths and perspectives of various contributors, and has also been shown to enhance creativity (Barrett et al., 2021).

Using LLMs as tools to assist humans with writing stories is an active research area (Chakrabarty et al., 2024b; Mirowski et al., 2023; Ippolito et al., 2022). In our experiments, stories are written exclusively by models without humans in the loop. However, our framework is fairly general allowing for human-machine collaboration at various stages of content creation.

**Multi-agent Systems** LLM-based agents have recently shown robust reasoning and planning capabilities across various application domains (Zhao et al., 2023; Bubeck et al., 2023). Multi-agent systems involve multiple independent LLMs working together to solve complex tasks that are beyond the capability of any individual agent (Talebirad & Nadiri, 2023; Park et al., 2023; Han et al., 2024; Guo et al., 2024). The agents are typically specialized in different aspects of a problem or have different roles, allowing the system to approach tasks in a more coordinated, distributed, and modular way. LLM-based multi-agent systems have recently demonstrated promising results in multiple areas including software development (Hong et al., 2024), robotic tasks such as motion planning (Mandi et al., 2024), simulations of human behavior (Park et al., 2023; Hua et al., 2024), the creation of gaming enviroments (Hu et al., 2024), recommender systems (Zhang et al., 2024a), simulations of financial trading (Li et al., 2023), and policy making (Xiao et al., 2023). We are not aware of existing multi-agent frameworks for long-form writing. We draw inspiration from related work demonstrating that collaborative problem-solving improves LLM task-solving capabilities (Hao et al., 2023; Wang et al., 2024; Zhang et al., 2024b). Our agents each adopt a specialized writing subtask and communicate through a shared scratchpad (or memory) which allows to effectively recall and utilize contextually-relevant past knowledge. In our experiments, we predefine the number and type of agents best suited to our story writing task, rather than dynamically generate agents based on story content (Chen et al., 2024).

**Evaluation** Story evaluation is admittedly a challenging task for humans and machines. Human evaluation is usually considered as the gold standard, but it is expensive, time-consuming (Guan & Huang, 2020), and can be subjective. It also cannot capture diversity since a model that copies directly from the training set would potentially pass the human quality bar without displaying any generalization or creativity (Hashimoto et al., 2019). Automated evaluation metrics based on lexical overlap or semantic similarity between generated stories and their human references have been shown to correlate poorly with human judgements (Chhun et al., 2022). In this paper, we introduce an LLM-based evaluator (Liusie et al., 2023; Liu et al., 2024; Zheng et al., 2024; Bohnet et al., 2024) to perform side-by-side comparisons of system outputs which correlates with human judgements. Inspired by recent proposals on how to assess human creativity (Chakrabarty et al., 2024a), we distill the story evaluation task into a few dimensions (e.g., plot, language use) which humans and machines can judge reliably.

## 3 AGENTS' ROOM

In this section, we formalize AGENTS' ROOM, the proposed multi-agent framework for collaborative writing. Given a complex writing task $x$, we generate output $y$, by decomposing the writing process into multiple subtasks tackled by specialized agents. The full AGENTS' ROOM framework is summarized in Algorithm 1 and explained below.

---

**Algorithm 1** AGENTS' ROOM framework

---

$s \leftarrow x$          $\triangleright$ Initialize the scratchpad
**while** $o(s, \mathcal{A}) ==$ True and $t < T$ **do**      $\triangleright$ While the orchestrator assigns a next agent
     $a_t = o(s, \mathcal{A})$      $\triangleright$ Select an agent given scratchpad
     $y_t = a_t(s)$      $\triangleright$ Obtain agent's output
     $s \leftarrow (s; (l_t, y_t))$      $\triangleright$ Update scratchpad
     **if** type$(a_t) ==$ writing **then**      $\triangleright$ If the agent is a writing agent, write to the final output
         $y \leftarrow (y; y_t)$
     **end if**
**end while**
**return** $y$      $\triangleright$ Return the final output

---

**Agents** We define an *agent* $a \in \mathcal{A}$ as a specialized model that takes text as input and returns text as output, specified by a unique identifier label $l$ and a mapping $f : \mathcal{V}^* \to \mathcal{V}^*$ (where $\mathcal{V}$ are vocabulary tokens). Each agent is specialized in a specific subtask. Under this definition, an agent can be a LLM fine-tuned for the subtask, a zero-shot prompted LLM with a specific input prompt, a deterministic text processing function (e.g., string formatting and parsing), or even a human interacting with the system. Herein, we focus on LLM-based agents, but we formalize the general framework's modeling assumptions (e.g., agent inputs and outputs as text instead of latent variables) to allow future work to incorporate human agents as well (e.g., by iteratively editing LLM-generated text). We define two types of agents (see below), namely *planning* and *writing* agents, which differ both in function and in their mode of interaction with the generated output.

**Multi-agent Communication** Communication between agents is critical for the successful completion of their tasks. While different forms of communication are possible, such as debate (Khan et al., 2024; Zhang et al., 2024b) or competition (Cheng et al., 2024), in this work we focus on *collaborative* communication since it would transfer most naturally to human-LLM collaborations. Collaborative agents work together towards a shared goal, exchanging information to enhance a collective solution.

**Scratchpad** The overall system requires a mechanism for sharing and managing information across the different agents. We assume our agents have access to a shared *scratchpad* $s \in \mathcal{V}^*$ that maintains individual agents' outputs and is passed along to the next agent. The scratchpad is initialized with the initial writing prompt $x$ and is then updated after each agent call. At each step $t$, the current agent $a_t$ takes as input the current scratchpad $s_t$ and generates output $y_t$. At each step, the scratchpad is updated with the agent's unique identifier and output such that $s_{t+1} \leftarrow (s_t; (l_t, y_t))$. We include the agent's label so that individual agents can easily reference and parse specific portions of the scratchpad to complete their subtask. Note that in this framework, the scratchpad does not contain the specific input prompt of a given LLM agent. Indeed, it is considered part of each agent's subtask to process the output $y_t$ into a suitable format to be used by other agents. Since agents have access to the scratchpad, this means that they can avoid writing redundant and duplicate information.

**Orchestrator** We have opted for a centralized architecture, where a central *orchestrator* determines the order upon which individual agents are called, and whether calling on each agent is necessary (e.g., depending on the task). Given a scratchpad $s_t$ and a set of available agents $\mathcal{A}$, the orchestrator $o : \mathcal{V}^* \times \mathcal{A}^* \to \mathcal{A}$ determines the next agent $a_{t+1}$ to call. It can be modeled as a Markov process, since each step depends entirely on the state of the scratchpad at step $t$. This orchestrator can be a discrete deterministic process, can have learnt transition probabilities, or can be arbitrarily complex. The orchestrator determines the stopping condition, i.e, when there is no more agent to call, or when a maximum number of steps $T$ has been reached.

**Planning Agents** Previous work (see Section 2) shows that LLMs benefit from an intermediate planning stage before generating the final output. These intermediate steps improve the overall output but are not included in the final output. We define *planning agents* as agents that specialize in generating these intermediate steps and write exclusively to the scratchpad. For instance, when writing a story, planning agents can draft character descriptions and plot elements; when writing an essay, they can outline the argumentative structure and retrieve references to substantiate claims. Since their outputs are in text format, a human-in-the-loop could review or edit these intermediate stages to guide the generative process.

**Writing Agents** Certain complex tasks, such as generating particularly long outputs or with sections written in different styles, remain challenging for a LLM to generate in one go. In such cases, the final output benefits from being generated section by section through separate agent calls. We define *writing agents* as agents specializing in writing specific parts of the final output. In addition to writing to the scratchpad, these writing agents iteratively write pieces of the final output $y$. Therefore, the final output can be formalized as the concatenation of the outputs of all the writing agents. For story writing, writing agents can specialize in specific parts of the narrative arc, such as the exposition or the climax; when writing an essay, they can each tackle different sections, such as the arguments in favor versus against.

## 4    FICTION WRITING TASK

In this section, we present an instantiation of the AGENTS' ROOM framework for fiction writing: given an initial writing prompt $x$, generate narrative $y$. We also introduce TELL ME A STORY[1], a new high-quality dataset of human-created writing prompts and fiction stories.

### 4.1    SPECIALIZED AGENTS INSPIRED BY NARRATIVE THEORY

We design specialized agents for our fiction writing task by drawing inspiration from narrative theory. We design four *planning agents*, each specialized in a specific aspect of story planning: [CONFLICT] defines the central conflict (e.g., a young boy has to fight an evil wizard who killed his parents), [CHARACTER] develops the character(s) (e.g., the young man is brave, caring, determined, loyal to his friends with a strong moral compass), [SETTING] develops the setting (e.g., most of the story takes places in the Hogwards School of Witchcraft and Wizardry, a fictional boarding school of magic for young wizards), and [PLOT] outlines the plot elements (e.g., the boy discovers he is the son of famous wizards and will attend Hogwarts School of Witchcraft and Wizardry). These planning agents target specific weaknesses observed in LLM-generated stories. Indeed, LLMs struggle writing compelling plots and consistent characters throughout long stories (see Section 2).

In addition to these content planning agents, we design five *writing agents*, each specialized in distinct elements of a typical story structure: [EXPOSITION], [RISING ACTION], [CLIMAX], [FALLING ACTION], and [RESOLUTION]. We adopt this structure since it is widely used and quite versatile (Freytag, 1896; Pavis, 1998), leaving other narrative structures for future work. When generating in one go, LLMs struggle to meet length requirements (e.g., specified in the prompt), resulting in stories that are generally too short (see Section 7). Since our writing agents generate the final output piecemeal, section by section, this results in longer outputs.

We model each of these agents as an LLM with a specific prompt template that formats the scratchpad into an appropriate prompt for each agent's subtask. Detailed scratchpad formatting and prompt templates for each agent are provided in Appendix B. To coordinate between the different agents, we define a deterministic *orchestrator* that first calls the planning agents as follows: [CONFLICT] → [CHARACTER] → [SETTING] → [PLOT], then calls the writing agents: [EXPOSITION] → [RISING ACTION] → [CLIMAX] → [FALLING ACTION] → [RESOLUTION]. We choose to use a deterministic orchestrator for simplicity, given the strong narrative theory prior. In future work, more refined orchestrators with learned objectives can be explored, possibly expanding to a wider range of narrative structures. As a first step towards building adequate reward models for training such orchestrators, we investigate automated evaluation strategies for the long-form fiction writing task in Section 6.

### 4.2    SYNTHETIC DATA GENERATION FOR AGENT TRAINING

For each agent, we explore zero-shot prompted and fine-tuned approaches, since a different degree of subtask specialization can be achieved through each approach. Fine-tuning requires agent outputs, which are not readily available; planning agent outputs such as plot and setting are usually not provided in datasets, while writing agent outputs require the stories to be split into its constituent parts. Similar to previous work (Schick et al., 2022; Narayan et al., 2023; Josifoski et al., 2023), we propose to generate synthetic outputs for these agents through distilled backtranslation.

Specifically, given as input writing prompts *and* stories from a dataset (see Section 4.3), we zero-shot prompt a larger teacher LLM to (1) generate the planning agent outputs, and (2) segment the

> **Example Prompts**
>
> Write a story about someone who is haunted by a ghost who wants to give business advice. This story should be around 2500 words. Don't make it scary. The main character is trying to make her food truck popular, so she travels around the southwestern part of the country in her food truck to gain more popularity. After a long time on the road, she comes home to find a ghost. This ghost doesn't want to scare her. He wants to give her business advice because he loved her food when he was alive. In the end, she accepts the help of the ghost.
>
> - - - - - - - - - - - - - - - - - - - - - - - - - - - - - - - - - - - - - - - - - - - - - - - - - - - - - - -
>
> Write a science fiction story about someone who is a time traveler and has dedicated everything in their life towards a goal, and now wonders if it was worth it. The story should be between 850 and 900 words. The story should begin with the main character waking up on a frozen tundra. He looks for shelter from the cold. He sees a dead wooly mammoth and realizes he traveled back to the ice age. The character should find shelter, and a predator is outside his shelter at night. The ending should not be happy.

Figure 2: Prompts from the TELL ME A STORY dataset (corresponding stories are in Appendix A).

Table 1: Comparison of TELL ME A STORY against existing open-ended story generation benchmarks. We report statistics on the number of training, validation, and testing instances; Input/Target denote the average number of tokens in the input (aka prompt) and target text.

| Dataset | number of examples | | | avg. token length | |
| --- | --- | --- | --- | --- | --- |
| | Training | Validation | Testing | Input | Target |
| WRITINGPROMPTS (Fan et al., 2019) | 272,600 | 15,620 | 15,138 | 28 | 735 |
| ROCSTORIES (Mostafazadeh et al., 2016) | 176,688 | 9,816 | 4,909 | 9 | 41 |
| CHANGEMYVIEW (Hua et al., 2019) | 42,462 | 6,480 | 7,562 | 18 | 104 |
| WIKIPLOTS[3] | 69,288 | 8,661 | 8,662 | 4 | 195 |
| **TELL ME A STORY** | 123 | 52 | 55 | 113 | 1,498 |

story into distinct parts (e.g., exposition, climax). Note that unlike typical distillation methods, our task is more straightforward; all that is required is to reverse engineer the agent outputs from an existing story rather than generate them from scratch. The teacher LLM outputs are then used to generate synthetic training datasets for both planning and writing agents. Detailed prompt templates are provided in Appendix C.

## 4.3 TELL ME A STORY DATASET

Creative writing presents a particular challenge from a data collection perspective; it is not akin to any traditional annotation or evaluation task where a requester provides some input and some set of guidelines for marking up that input in a consistent manner. While standards exist for "good" writing, they evaluate the quality of writing across multiple interdependent and independent dimensions, all at once. In addition to this, the skill of writing really represents several skills that are learned over the course of a person's lifetime and educational experience. Furthermore, evaluating writing necessarily requires the subjective stance of the evaluator.

Taking into consideration all of these complexities, we collected TELL ME A STORY through writing workshops to replicate the organic environment in which a collaborative writing process can take place. We provided a group of writers (28 in total) with broad instructions for quality based on collation of the Publication Manual of the American Psychological Association (currently in its 7th edition), the GRE/GMAT writing assessment rubrics, and various mass market style guides. Writers created their own prompts, wrote an initial draft, received feedback from peers, revised, and then submitted to a workshop lead for a second round of feedback and final approval. Workshop leads could ask for additional edits or accept as is. Workshops lasted on average 3–4 weeks. option of downtime between workshops or the opportunity to work in another workshop if they desired to prevent burnout. The average rate of production for workshops generally reached no more than 2–3 writing samples per writer per week. We provide example prompts in Figure 2 and example stories in Appendix A. The majority of the stories belong to the genres of science fiction and fantasy but are also representative of the following genres: horror, drama, comedy, adventure, and folklore.

---

[3]Available at: `https://github.com/markriedl/WikiPlots`

Table 1 compares TELL ME A STORY against commonly used story generation benchmarks. Our dataset is small in scale and thus not suited to training a model from scratch. Our prompts are more detailed compared to other benchmarks (see Input column) and the target stories are genuinely long (e.g, double in size compared to WRITINGPROMPTS). Note that some of these datasets, although useful for system development, are not strictly speaking narratives. WIKIPLOTS is a collection of plots from Wikipedia rather than stories, ROCSTORIES are five-sentence long common sense stories, and CHANGEMYVIEW contains pairs of posts and their counter-arguments.

## 5 EXPERIMENTAL SETUP

**Comparison Systems** The state-of-the-art approach for generating narratives consists of generating the story in one go, either through zero-shot prompting (see Figure 2) or fine-tuning, which we denote as $E2E_{ZS}$ and $E2E_{FT}$, respectively. We also experimented with more detailed instantiations of $E2E_{ZS}$ by instructing the model to: (1) generate the central conflict, characters, setting, and plot before generating the story ($E2E_{ZS}$ plan); (2) reflect on the central conflict, characters, setting, and plot according to detailed guidelines, before generating the story ($E2E_{ZS}$ reflect); self-reflection (Madaan et al., 2023; Shinn et al., 2023) has been previously explored to solve intricate tasks that could be challenging for zero-shot prompting; both baselines ($E2E_{ZS}$ plan, $E2E_{ZS}$ reflect) use the same detailed instructions provided to our planning agents (see Appendix B.2); (3) generate a plan automatically before generating the story in one call ($E2E_{ZS}$ decompose); in this case, plans are predicted without any task-specific knowledge, story generation is decomposed into a series of simpler sub-problems which are solved sequentially (Yang et al., 2022; Khot et al., 2023); and (4) generate a plan for the story first followed by a second call in which the model is instructed to generate the story based on the input prompt and the plan (2STAGE decompose).

**AGENTS' ROOM Variants** We use the *plan+write* tag to denote the AGENTS' ROOM variant with the writing and planning agents as previously described (see Section 4.1). To explore trade-offs between the different types of agents, we investigate two additional variants, *plan* and *write*, where we use only planning, or only writing agents, respectively. In the specific case of the *plan* variant with only planning agents, we still need a writing agent to finalize the story, since planning alone does not result in a final story. Therefore, the *plan* variant includes a single simple writing agent, which we denote as the [FINALIZER]. The prompt template for the [FINALIZER] agent is provided in Appendix B. We investigate both zero-shot and fine-tuned agents. For each AGENTS' ROOM variant, we explore two settings, one with only zero-shot agents, and one with only fine-tuned agents, denoted as $AR_{ZS}$ and $AR_{FT}$, respectively. Since agents are called independently, it is possible to mix and match between zero-shot and fine-tuned agents, but we keep the two settings separate to derive clearer signal for each approach.

**Implementation** For all comparison baselines and AGENTS' ROOM agents, we use a Gemini 1.5 Flash[4] backbone, a lightweight and cost-efficient model that has demonstrated good performance across a wide range of generative tasks (Reid et al., 2024). In particular, it features long context capabilities (up to one-million tokens) which makes it suitable for handling the scratchpad with multiple agents' contributions. We use input length out of {1,024, 2,048, 4,096, 8,192} tokens depending on the length of the scratchpad and a target token length of 4,096. While the outputs generated by the baseline systems are generally shorter than what is requested in the original prompt (see Section 7), we observe no improvements when increasing the target token length. We hypothesize that the observed limits on output lengths are likely due to the backbone model being trained on data with mostly shorter outputs.

For the synthetic training data generation described in Section 4.2, we use Gemini Ultra[4] (Team et al., 2023) as the teacher model. Since our dataset contains only a small number of training examples, we fine-tune our models ($E2E_{FT}$ and individual agents for $AR_{FT}$) using LoRA (Hu et al., 2021), a computationally-efficient approach that updates only a small portion of the model weights. We perform LoRA-tuning with rank 4 and a learning rate of $1^{-6}$ (picked after a hyperparameter search through $\{1^{-4}, 1^{-5}, 1^{-6}, 1^{-7}\}$). We LoRA-tune for 250 steps with a batch size of 16, saving checkpoints every 20 steps. We then select the checkpoint with lowest loss on the validation set.

---

[4]Available at: `https://cloud.google.com/apis`

# 6 EVALUATION

We evaluate the quality of the generated outputs along several dimensions through human judgment elicitation and automated evaluation methods.

## 6.1 HUMAN EVALUATION

We evaluate system output by soliciting pairwise preferences (Louviere & Woodworth, 1990) along four dimensions, as well as an overall preference. We distill previous proposals (Chakrabarty et al., 2024b; Chhun et al., 2022) on how to evaluate creative writing into the following criteria:

- **Plot** — Does the story have a recognizable structure, e.g., with a connected beginning, middle, and end? Does it exhibit events and turns that move the plot forward without logical or conceptual inconsistencies?

- **Creativity** — Does the story have engaging characters, themes, and imagery? Does it avoid overly cliched characters and storylines, unintentional tropes, and stereotypes? Does it include original elements that were not explicitly mentioned in the prompt?

- **Development** — Are the characters and settings contextualized with relevant details that allow the reader to understand their place in the story? Are appropriate levels of detail and complexity provided to lend the story a feeling of realness and believability?

- **Language Use** — Does the language used feel varied and rich? Does the story exhibit rhetorical, linguistic and literary devices to create interesting effects? Does it avoid bland or repetitive phrases?

The full instructions are reproduced in Appendix D. Participants are shown two stories and asked to decide which one is better in each dimension. They can also respond that the two stories are about the same. Participants are allowed to rate up to five samples in one sitting, due to our task being cognitively taxing and time-consuming. We assign samples to participants following a Latin Square design, such that each participant does not rate the same writing prompt more than once. We randomize the order in which the two stories are shown to mitigate presentation order as a potential bias. We gather ratings for all examples included in the TELL ME A STORY test set and compare outputs from all E2E and AGENTS' ROOM variants (see Figure 3); we also include the human-written stories as an upper bound. Our annotators were writers or had a degree in related disciplines (e.g., literature). We obtained a total of 9,900 pairwise ratings which we converted into systems' relative strengths using a Bradley-Terry model (Bradley & Terry 1952; see Section 6.2). Inter-annotator agreement was $\kappa = 0.46$ ($p < 0.01$, $N = 150$, $k = 3$), as measured by Fleiss' Kappa, which we interpret to be satisfactory given the subjectivity of our task.

## 6.2 AUTOMATIC EVALUATION

Many previous studies (see Yang & Jin 2024 and the references therein) have highlighted the challenges associated with evaluating narratives automatically. Metrics based on lexical matching correlate poorly with human judgments (Chhun et al., 2022; Chakrabarty et al., 2024a) and do not effectively measure story quality (e.g., is the story original and well-written with plausible characters). In this work, we report reference-based metrics, specifically Rouge-L (Lin, 2004) and BertScore (Zhang et al., 2020), but also adopt several surface-based metrics[1] intended to capture differences between human writing and LLM-generated stories. Specifically, we compute *story length* to determine whether models are able to generate long stories and quantify *structural differences* between human and machine stories (e.g., number of sentences starting with an article or a pronoun). We also measure the ratio of *unique words* in a story which gives an idea of creative language use, and *intra-* and *inter-story trigram repetition* (Yao et al., 2019; Goldfarb-Tarrant et al., 2020) which capture diversity within a story and across stories (high inter-story repetition suggests models generate similar stories even when given different prompts). Finally, trigram *overlap with the prompt* is used to indicate whether models can creatively elaborate on the information provided.

In addition, we develop a LLM-based evaluator (Liusie et al., 2023; Liu et al., 2024; Zheng et al., 2024; Bohnet et al., 2024) to perform *side-by-side* comparisons of system output. We design prompts targeting the same dimensions of story quality adopted in our human evaluation. Specifically, we

Table 2: Comparison between human and model generated stories using automatic metrics (TELL ME A STORY test set): #words (average number of words per story), #para (average number of paragraphs per story), Article (proportion of sentences starting with an article), Pro (proportion of sentences starting with a pronoun), Unique (percentage of unique words), Intra (intra-story trigram repetition), Inter (inter-story trigram repetition), Overlap (proportion of trigrams overlapping with the prompt). We also report two reference-based metrics, Rouge-L and BertScore. AR abbreviates AGENTS' ROOM systems; subscripts $_{ZS}$ and $_{FT}$ respectively refer to zero-shot and fine-tuned.

| Models | #words | #para | Article | Pro | Unique | Intra | Inter | Overlap | Rouge | BertSc |
|---|---|---|---|---|---|---|---|---|---|---|
| E2E$_{ZS}$ | 1,207 | 32.24 | 12.74 | 40.45 | 44.57 | 28.78 | 33.35 | .0034 | 20.71 | .8152 |
| E2E$_{ZS}$ plan | 1,130 | 27.24 | 15.34 | 42.25 | 45.93 | 23.59 | 29.41 | .0027 | 20.58 | .8173 |
| E2E$_{ZS}$ reflect | 1,126 | 28.62 | 13.79 | 40.96 | 45.85 | 23.68 | 23.95 | .0032 | 20.36 | .8152 |
| E2E$_{ZS}$ decompose | 965 | 21.25 | 21.36 | 39.62 | 45.49 | 31.98 | 44.10 | .0034 | 19.41 | .8067 |
| E2E$_{FT}$ | 1,193 | 32.25 | 12.58 | 43.39 | 44.02 | 28.21 | 31.31 | .0036 | 20.73 | .8138 |
| 2STAGE decompose | 1,090 | 24.82 | 15.59 | 42.15 | 44.50 | 21.54 | 24.26 | .0031 | 20.35 | .816 |
| AR$_{ZS}$ plan | 926 | 20.95 | 13.82 | 40.68 | 43.88 | 29.70 | 33.49 | .0017 | 19.58 | .8119 |
| AR$_{ZS}$ write | 3,278 | 63.80 | 25.32 | 39.91 | 34.97 | 47.50 | 44.09 | .0022 | 17.34 | .8103 |
| AR$_{ZS}$ plan + write | 3,034 | 58.65 | 15.97 | 41.43 | 35.05 | 44.73 | 43.25 | .0022 | 17.57 | .8123 |
| AR$_{FT}$ plan | 856 | 21.05 | 18.02 | 39.29 | 44.65 | 23.85 | 28.05 | .0027 | 19.24 | .8146 |
| AR$_{FT}$ write | 3,129 | 61.90 | 17.45 | 44.80 | 36.35 | 46.39 | 42.39 | .0021 | 17.53 | .8150 |
| AR$_{FT}$ plan + write | 3,006 | 56.85 | 17.52 | 43.03 | 34.30 | 46.31 | 41.60 | .0019 | 17.60 | .8152 |
| Humans | 1,439 | 32.91 | 10.01 | 32.37 | 50.35 | 15.53 | 19.24 | .0020 | — | — |

adapt the evaluation criteria described in Section 6.1 into a prompt template shown in Appendix E. This template asks the evaluator for a detailed assessment of the two stories presented, followed by a final conclusion, which is then parsed to obtain preference scores for each dimension. We provide an example usage in Appendix E. Given $N$ system outputs for each of $M$ input prompts, we evaluate all possible (unordered) pairs of outputs for each input (while shuffling the order in which the outputs are presented), producing $M \times N \times (N - 1)/2$ different pairwise ratings. Finally, we obtain a wins matrix $W$ where $w_{i,j}$ is the number of times system $i$ wins over system $j$. This matrix is then used to obtain the systems' relative strengths after fitting a Bradley-Terry model (Bradley & Terry, 1952). We use Gemini 1.5 Pro[4] as our LLM evaluator, as suggested in Bohnet et al. (2024).

## 7 RESULTS

Table 2 compares human and model generated stories using surface- and reference-based metrics. As far as story length is concerned, we observe that E2E stories are slightly shorter than human ones, while planning models are shortest overall. However, models which include writing agents produce considerably longer stories (by a factor of two) with more dialogue as suggested by the increased number of paragraphs. We also find machine stories to be more generic in their sentence structure as evidenced by the higher proportion of stories which start with an article or pronoun. Human-written stories are also more diverse (less repetitive) as shown by the higher ratio of unique words and less repeated trigrams (Inter and Intra in Table 2). The most repetitive models are also the ones that produce the longest stories. In terms of overlap with the prompt, we find AGENTS' ROOM systems to copy least, at a rate similar to that of human writers. Rouge-L rewards the E2E systems most, as they least deviate from the prompt gold standard stories, while BertScore is not very discriminating, equally preferring the simplest (E2E$_{ZS}$) *and* most complicated system (AR$_{FT}$ *plan+write*). Examples of stories written by humans and machines can be found in Appendix A.

Figure 3 reports system rankings obtained from human judgments and the LLM evaluator. For the sake of brevity, we omit baselines shown to underperform in Table 2 (i.e., E2E$_{ZS}$ with *plan, deflect, decompose* and 2STAGE *decompose*) but include these results in Appendix F.

**Human-written stories are preferred overall** As shown in Figure 3a, human judgments reveal a performance gap between machine writers (E2E and AGENTS' ROOM) and professional writers, a finding that is in line with Chakrabarty et al. (2024a). We observe this gap across all dimensions, but

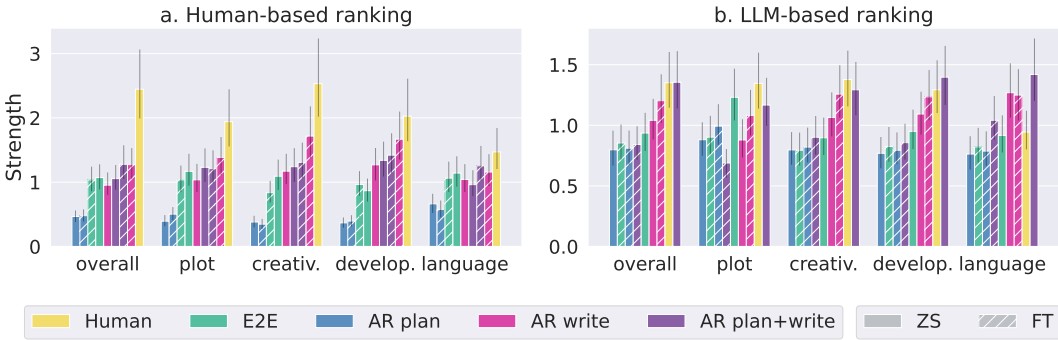

Figure 3: Overall system ranking across dimensions of plot, creativity, development, and language, according to human ratings (a) and a LLM-based evaluator (b).

we note that it is smaller in the language use dimension. This result suggests that while machine-generated stories still fall short in terms of compelling plots and unique ideas, LLMs, in their current state, may be useful as writing assistants. To ensure that the preference towards human stories is not merely due to them being longer, we computed the proportion of pairwise comparisons for which our human raters preferred the longer story overall (excluding ties) and found it to be around 0.51.

**AGENTS' ROOM outperforms baseline systems** Across all dimensions, our participants prefer AGENTS' ROOM stories with writing agents over those produced by baseline systems, with the $AR_{FT}$ *write* and $AR_{FT}$ *plan+write* systems performing best. Aside from rating the stories, participants had the option to leave feedback on their quality; we provide samples of this feedback in Appendix F. AR *plan* variants do not perform that well, most likely due to the single [FINALIZER] agent being too simplistic to make good use of the planned elements provided in the scratchpad. We note that fine-tuned agents yield better results over zero-shot ones, which shows that generating synthetic data by back-translating from gold standard outputs (see Section 4.2) is an effective strategy for training specialized agents for different subtasks. Finally, we observe similar trends with smaller scale models (see Appendix F for additional results).

**The LLM evaluator agrees with humans and itself** The LLM-based rankings in Figure 3b reveal similar tendencies to human ratings. The LLM overall prefers human stories and those generated by the AR *plan + write* system against all other model variants, even though it does not discriminate very strongly between those two. LLM-based judgments of story quality correlate significantly with human ratings across all dimensions, both by systems (Spearman's rank correlation $\rho = 0.62$; $p < 0.01$, $N = 45$) and by items ($\rho = 0.41$; $p < 0.01$, $N = 9,900$). The LLM and humans have the highest agreement when assessing story development ($\rho = 0.83$, $p < 0.01$) and creativity ($\rho = 0.85$, $p < 0.01$). Similarly to the findings in Bohnet et al. (2024), we also find that the LLM evaluator scores are consistent: 90.2% of the time the LLM prefers the same story in a second run, when the stories are presented in the opposite order.

## 8 CONCLUSION

We propose AGENTS' ROOM, a general framework for multi-agent collaborative writing, and describe its instantiation for the long-form fiction writing task. Drawing inspiration from narrative theory, we decompose the complex writing task into subtasks tackled by specialized agents. To illustrate this framework, we present TELL ME A STORY, a high-quality dataset of prompts and long-form stories collected through multiple rounds of writing workshops with human participants. We show that AGENTS' ROOM generates stories that are preferred by human evaluators over those produced by baseline systems. Moreover, we demonstrate effective training stategies for developing specialized agents by leveraging synthetically-generated data. We introduce a human evaluation framework for evaluating long-form narratives across several dimensions, and an LLM-based evaluator that correlates significantly with human raters. With improved automated evaluation, future work can explore more sophisticated orchestrators, including the development of reward models and learning objectives for such orchestrators.

ETHICS STATEMENT

There are a number of ethical considerations when using generative language models. While the work we present here makes a step towards improving the quality of text generation systems, it is important to note that current systems are still far from perfect in this respect and may make mistakes. In particular, generative models may perpetuate biases present in their training data. Even when writing fiction, the models may inadvertently amplify societal biases or reinforce stereotypes, leading to the production of biased content. Therefore, the generated outputs should be meticulously verified and used with caution.

REPRODUCIBILITY STATEMENT

For reproducibility, we release the TELL ME A STORY dataset on which we conduct our experiments, complete with its train, validation, and test splits, as described in Section 4.3. We specify the model backbones, implementation details, and where to access the checkpoints in Section 5. All prompt templates and scratchpad formatting templates are provided in Appendix. For the evaluation, we provide the exact rater instructions in Appendix D and the LLM evaluator prompts in Appendix E.

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

## A  Examples from Tell me a story Dataset and System Output

**Example Prompt**

Write a story about someone who is haunted by a ghost who wants to give business advice. This story should be around 2500 words. Don't make it scary. The main character is trying to make her food truck popular, so she travels around the southwestern part of the country in her food truck to gain more popularity. After a long time on the road, she comes home to find a ghost. This ghost doesn't want to scare her. He wants to give her business advice because he loved her food when he was alive. In the end, she accepts the help of the ghost.

**Human-written Story**

"Finally, I can sleep in my own bed for once," Rita said.

She plopped down on her bed in an attempt to soak in its comfiness. But then there was a knock at the door, in her mind she said "Why!"

Rita dragged herself from her bed to check to see who it was, but when she got to the door, there was no one there. She looked around to see if anyone was around, but there was nothing. She wondered if anyone with kids had moved in while she was away.

Disregarding the thought she went back into her apartment and went back to her bed. She didn't care to remove her shoes or her coat, that was just how tired she was. Rita had been on the road for the past five months trying to drum up enough money and support to start her own restaurant. Currently, she had been operating out of a food truck that she had started a year ago. She went from city to city every week, trying to find the perfect place that garnered enough foot traffic and business.

After five months of going around the southwestern part of the United States, driving, cooking, serving, and advertising her business wherever she went, she decided that it was time to go back to home base and rest up a bit before she went back to the grind. It was three o'clock in the morning when she finally got up to get a little more comfy.

She went into the kitchen where she had left her bags since it was the first place you reached once you entered the two-bedroom apartment. Rita didn't have the energy to drag them along with herself to her room at the back of the apartment.

Once she got to the kitchen she noticed that the flyers she had made for her food truck were scattered on the ground. She picked them up, but what she found on the flyers when she looked at them scared her.

Rita could have sworn she saw something moving in the flyer; she admitted the flyers had an intricate design that looked like an optical illusion but what she saw was not a part of the design. Taking a second look, she did not see what she thought she saw.

"Nope nope nope, didn't see a thing." Rita decided that she just needed more rest after all it was three in the morning. Back in her room, Rita settled to get back in bed when she heard the knocking, but this time it was coming from her bathroom. She was definitely freaking out now. She grabbed the closest thing she could use as a weapon. A wooden crate, with a faded beer label on it, was all there was in her line of sight.

She tiptoed her way to the bathroom, with the crate held above her head ready to strike anything that moves. She pushed the door open with her foot, and saw a man standing in her bathroom. She swung the crate, but it passed through the man, and she fell backwards. Seeing this, she wanted to close the door, but she couldn't since she had already fallen to the floor and was backing up from the entrance to the bathroom. Gathering the courage to approach the bathroom again, Rita saw nothing but a note on her cheval mirror that read, "Please don't be afraid of me, I just want to help you." The bathroom was ice-cold all of a sudden. Still standing in the doorway, Rita saw the man again, and this time he waved. After the initial shock of seeing him the first time she just waved back.

Taking a deep breath in and exhaling, Rita said, "Okay, how is it that you want to help me?" She thought she was going crazy or something because she intently waited for this transparent being to communicate with her. She refused to admit it was a ghost, so she just waited.

Finally, the being began writing on the mirror again. It was freezing cold in the bathroom so the mirror appeared to have frost on it. He explained in writing, "this would go faster if you would allow me near you."

Reluctantly, she agreed, the man approached her, and she decided to close her eyes as if she could somehow pretend this wasn't happening. He touched her shoulder, all of a sudden Rita was hearing him in her mind.

"Hi, I'm Jeff. I must say your cooking was the absolute best when I was alive." Shocked by his admission, Rita opened her eyes to look at him.

Jeff stood there with a smile on his face and continued, "I had the pleasure of trying your southern fried cabbage the day I died. It was truly the most fitting last meal." Amazed by his candor, Rita tried to remember when she may have seen this guy at one of the many locations she had been to in the last five months, but she couldn't place him. Giving her time to digest the information, Jeff continued, "You won't remember me because I wasn't the one that picked up the order, it was my wife. But anyway, let's get back to the point here. I want to help you with your food truck venture." Rita took a minute and thought, "How would a dead man help me with this?" Jeff smirked at her because he knew she was thinking this was crazy. She shrugged her shoulders and gave in, asking, "Okay, how do you suppose you are going to help, because I don't think having a ghost doing tricks will attract people."

Jeff shook his head and just looked at Rita until she finished her muttering.

"First and foremost, I don't do tricks. Second, you need an upgrade girlie girl. Who uses printed out flyers anymore in this technological world? Also, that little design you got on those flyers — scrap it. I got lost for hours in that thing, trying to get your attention."

Before Jeff could finish, Rita muttered under her breath, "I knew I saw something on those flyers."

Jeff continued to speak, "Rita, you need to expand even more than what you have. I'm not saying to expand your territory. I'm saying to reach more people you need to get on those food delivery apps, or at least create a website that will allow people to place orders." Rita listened on, and she knew what Jeff was saying was true. She was stuck in a time long past when technology was not as widespread.

After Jeff finished what he had to say, Rita asked, "So, if I do all of this, will you let me sleep in peace and stop knocking on everything to get my attention? If you have something to say just leave a note on the mirror."

Laughing at Rita's acceptance of him and the things he set out for her to do. He gave her some people to contact about setting up some things as far as the delivery service went. Then he allowed her to go to sleep.

"See you later Jeff, we will continue this when I wake up." Rita said as she closed the door to her room and finally got into her bed and fell asleep.

**System output:** E2E$_{ZS}$

The desert sun beat down on Amelia's food truck, "The Wandering Wok," as she flipped a sizzling stack of tofu, her brow furrowed in concentration. For the past year, she'd been crisscrossing the Southwest, her truck a mobile testament to her culinary dreams. She'd started with a simple concept: innovative vegetarian takes on classic Asian dishes, fueled by fresh, local ingredients. It was a hit in her hometown, but she craved more. Now, she was pushing the limits, trying new recipes, adjusting her menu, even experimenting with different locations.

Still, the crowds weren't as big as she hoped. Sometimes, there were just a few regulars, their faces familiar and comforting, but not enough to sustain her. The loneliness of the road was starting to get to her. The constant hustle, the uncertainty, the lack of a real home – it was wearing her down.

One evening, after a particularly disappointing day in Sedona, Amelia packed up her truck, feeling a familiar pang of disappointment. The drive back to her hometown in Arizona was long and quiet. By the time she reached her childhood home, a cozy bungalow with a sprawling backyard, she was exhausted.

Stepping inside, Amelia was greeted by the comforting scent of dust and memories. Her parents had passed years ago, leaving the house to her. She'd kept it as a haven, a place to return to when the road got too much. As she unpacked her bags, a strange sensation washed over her. It wasn't fear, but a sense of. . . presence.

A shiver ran down her spine as she turned, her gaze falling on a figure standing in the kitchen doorway. It was a man, tall and wiry, with a mischievous glint in his eyes. He was transparent, a faint shimmer of light outlining his form. A ghost.

Amelia's initial fear quickly dissipated, replaced by a strange sense of calm. The ghost, dressed in a faded, old-fashioned suit, smiled warmly. "Don't be alarmed," he said, his voice a gentle whisper. "I'm not here to scare you."

His words, though spoken in a spectral voice, held a reassuring warmth. Amelia, still reeling from the shock, simply stared at him.

"I'm Jacob," the ghost continued, his smile widening. "And I've been watching you."

Amelia's eyebrows shot up. "Watching me?"

"Yes," Jacob chuckled. "Your truck, 'The Wandering Wok.' I loved your food. I used to come by every week when you were still in town."

He gestured toward the kitchen table, where a plate sat, adorned with a single, perfectly-formed veggie spring roll. It was a dish Amelia had perfected years ago.

"I've been following you all over the Southwest, you know," Jacob said, his voice tinged with a hint of pride. "I've seen your struggles, your successes, your. . . lack of customers."

Amelia, still processing the situation, felt a blush creep up her cheeks. "Well, I'm trying," she mumbled, her voice barely above a whisper.

"You're trying too hard," Jacob said, shaking his head gently. "You're trying to be everything to everyone. You need to focus, find your niche."

His words hit Amelia like a jolt of electricity. It was true. She'd been trying to please everyone, offering everything from vegan pho to spicy Korean bibimbap, hoping to attract a wider audience. But in the process, she'd lost her own unique voice.

"You have a gift, Amelia," Jacob continued, his gaze intense. "You make food that's both delicious and beautiful. But you're hiding it behind a facade of. . . well, too much."

Amelia felt a wave of frustration wash over her. "It's not that easy," she said, her voice tight. "Competition is fierce. Everyone's trying to be the next big thing."

Jacob chuckled, a soft, ethereal sound. "Competition is good, Amelia. It pushes you to be better. But you can't compete if you don't know what you stand for."

He paused, his eyes twinkling. "You need to find your story, Amelia. What makes you different? What makes your food special?"

Amelia thought about it. Jacob was right. She'd been so focused on the bigger picture, on the grand vision of a successful food truck empire, that she'd forgotten the simple joy of creating.

"My grandma," she said, the words coming out in a rush. "She taught me how to cook. Her recipes, her stories, her love for food. . . that's what makes my food special."

Jacob smiled. "See? You have a story. A beautiful, personal story that people will connect with. You just need to tell it."

He went on, sharing his own story, a tale of a young man who had come to Arizona from a small town in the Midwest, seeking his fortune. He'd worked hard, saving up every penny to open a small diner, a place where he could share his love for food with the community. He'd been happy, he said, until a sudden illness had taken him away.

"I never got to share my dream with the world," Jacob confessed, his voice tinged with sadness. "But you, Amelia, you can. You have the talent, the passion, the story. All you need is to find your own path."

He looked at her, his expression serious. "You need to create a menu that reflects your heritage, your grandma's recipes, your own unique twist. You need to tell your story, Amelia. And you need to do it with passion, with love, with the same dedication your grandma had."

Amelia felt a surge of inspiration. Jacob's words, though spoken by a ghost, resonated deeply within her. She'd been so lost in the pursuit of success that she'd forgotten the essence of what made her food special.

"Thank you," Amelia whispered, her voice choked with emotion. "I. . . I think I know what I need to do."

Jacob smiled, a gentle glow radiating from him. "I knew you would," he said. "Now go, Amelia. Go out there and make your grandma proud."

As quickly as he had appeared, Jacob vanished, leaving Amelia alone in the quiet kitchen. The plate with the spring roll was still on the table, a testament to his presence, his love for her food.

Amelia spent the next few weeks reworking her menu. She incorporated her grandma's recipes, her own personal touches, her love for fresh, local ingredients. She created a menu that told a story, a story of family, tradition, and passion.

She also decided to rebrand her truck. "The Wandering Wok" was gone, replaced by "Grandma's Kitchen," a name that resonated with warmth and authenticity.

The first time she took her revamped truck out, Amelia felt a nervous excitement mixed with a sense of peace. She had found her voice, her story. As she set up her truck in a bustling market in Phoenix, she felt a familiar presence, a gentle warmth surrounding her. Jacob was there, a silent observer, his form shimmering faintly in the afternoon sun. He smiled at her, his eyes filled with pride.

"This is your story, Amelia," he whispered. "Tell it with your heart, and the world will listen."

Amelia took a deep breath, her heart filled with gratitude. She had found her path, her purpose, thanks to a ghost who simply wanted to share his love for food and his belief in her. And as she started cooking, her hands moving with newfound confidence, she knew that Jacob would be with her, cheering her on, every step of the way.

The crowds came, drawn in by the aroma of her grandma's recipes, the warmth of her story, the passion that radiated from her every move. They tasted the love, the care, the heritage woven into each dish.

Amelia's journey had taken a detour, a detour that led her back to her roots, to her grandma's kitchen, to the true essence of her culinary dreams. And as she saw the smiles on the faces of her customers, she knew that Jacob, the ghost with a passion for food, was finally at peace. His story, his dream, was living on, through her.

**System output: AGENTS' ROOM**$_{ZS}$

The desert sun beat down on the cherry-red food truck, baking the metal siding until it shimmered like a mirage. Lily, her forehead creased with a mixture of sweat and frustration, stared out at the empty parking lot, her reflection wavering in the heat haze. Another day, another disappointing turnout. The farmers' market had been slow, and the lunchtime crowd had been nonexistent. The southwest was vast and beautiful, but the road to success seemed endlessly long.

Lily sighed, running a hand through her messy brown hair. She'd dreamt of owning a food truck since she was a kid, watching her grandmother whip up culinary magic in their tiny kitchen. The smells of her grandmother's cooking, the warmth of the family gathered around the table, the joy in her grandmother's eyes as she presented each dish...it was a memory Lily cherished. She wanted to share that same joy, that same passion, with the world. But the reality was far from the dream. The food truck, her pride and joy, was becoming a burden. The money was dwindling, and the pressure was mounting.

She'd spent the last six months crisscrossing the southwest, driving from one bustling market to another, her truck a beacon of hope in a sea of uncertainty. She'd tried everything – catchy slogans, bright signage, even offering free samples. But nothing seemed to work. The competition was fierce, and the customers were fickle. Lily felt like she was losing her grip, her confidence slowly slipping away.

Exhausted and discouraged, she decided to head home. Her childhood home, nestled in a dusty corner of the small town of Tombstone, Arizona, was a place of comfort and memories. Maybe a change of scenery, a little time away from the relentless grind of the road, would help her clear her head. She needed a break, a chance to regroup and re-evaluate her strategy.

The drive back was quiet, the only sound the hum of the engine and the occasional sigh of the wind whipping through the open windows. As she passed through the familiar landscape, the desert wildflowers swaying in the breeze, the towering red rock formations casting long shadows across the parched earth, a sense of melancholy settled over her. The southwest, once a symbol of adventure and opportunity, now felt like a reminder of her own failures.

The sun had begun its descent, painting the sky with shades of orange and purple, as Lily finally turned onto the familiar street leading to her childhood home. The adobe house, with its terracotta tile roof and sun-baked walls, stood bathed in the golden light, a testament to time and resilience. It was a place of warmth and comfort, a place where she felt safe and loved. But as she pulled into the driveway, a shiver ran down her spine. A sense of unease, a feeling of being watched, crept into her heart. It was as if the house itself was holding its breath, waiting for something.

Lily shook her head, trying to dismiss the feeling. It was just her imagination, she told herself. She was tired, stressed, and probably just a little bit paranoid. But as she stepped out of the truck, the feeling intensified. The air around her seemed to crackle with an unseen energy, and a faint, cold breeze sent a shiver down her spine. She glanced around, but saw nothing. The street was empty, the only sound the chirping of crickets and the distant howling of coyotes.

She quickly shrugged off the unsettling feeling, grabbing her bags and heading towards the house. The front door, a heavy wooden structure adorned with a rusty brass knocker, creaked open as she pushed it, revealing a dark and dusty interior. The scent of old wood and forgotten memories hung in the air, thick and heavy.

As she stepped inside, a sense of unease settled over her. It wasn't just the dust and the shadows that made her uneasy, but an almost tangible feeling of presence, a sense of being watched. She switched on the light, illuminating the worn-out furniture and the faded photographs on the walls. The house, once a haven of childhood joy, now felt strangely empty, as if its heart had been ripped out.

Lily took a deep breath, trying to calm her nerves. She was home, and that was all that mattered. She would unpack, make a cup of tea, and then try to figure out her next move. She needed a plan, a way to get her food truck back on track. But as she started to unpack, a faint shimmer caught her eye. A flicker of movement in the corner of the room, barely visible, like a heat wave shimmering in the summer sun. Lily froze, her heart pounding in her chest. She blinked, rubbing her eyes, but the flicker remained, growing stronger, taking shape.

It was a man, a tall, slender figure with a kind face and a gentle smile. But he wasn't solid. He was translucent, like a wisp of smoke, and his clothes were faded and worn, as if he belonged to a different time. He stood there, watching her, his eyes filled with a mixture of curiosity and concern.

Lily's breath caught in her throat. Her mind raced, trying to make sense of what she was seeing. This couldn't be real, she told herself. She was exhausted, stressed, and probably hallucinating. But the figure didn't disappear. He simply stood there, watching her, his presence as real as the dust motes dancing in the sunlight streaming through the window.

And then, he spoke, his voice a gentle baritone, as comforting as a warm fire on a cold night. "Welcome home, Lily."

Lily's voice caught in her throat. She couldn't speak, couldn't even breathe. The man, the ghost, stood there, his translucent form shimmering in the afternoon light. He seemed to be smiling, his eyes crinkling at the corners as if in amusement, but the sight of him sent chills down her spine.

"Don't be afraid," the ghost said, his voice a gentle murmur. "I mean you no harm."

Lily managed to swallow, her throat dry as desert sand. "Who...who are you?" she stammered, her voice barely a whisper.

"My name is Mr. Henderson," the ghost replied, his voice a comforting baritone. "I used to live here, in this very house."

"You...used to live here?" Lily repeated, her mind struggling to grasp the impossible. "But...you're...you're a ghost."

Mr. Henderson chuckled softly, a sound like wind chimes tinkling in the breeze. "Yes, I am. But don't be afraid. I'm not here to scare you. I'm here to help."

Lily stared at him, her confusion growing. "Help?" she echoed, her voice filled with disbelief. "Help with what?"

"With your food truck," Mr. Henderson said, his smile widening. "I heard you've been struggling. I know a thing or two about cooking, you see. I used to be a chef, back in the day."

Lily's eyebrows shot up in surprise. "A chef?" she repeated, her skepticism battling with a sliver of curiosity. "But...how can you help? You're...you're a ghost."

Mr. Henderson chuckled again, his ghostly form shimmering faintly. "I may be a ghost, but I still have a passion for food. And I have a lot of experience. Perhaps I can offer some advice, some insights, that might help you on your journey."

Lily's mind whirled. The idea of a ghost offering business advice was absurd, preposterous even. But there was something about Mr. Henderson, something in his gentle demeanor, his genuine concern, that made her hesitant to dismiss him outright. She'd been struggling, feeling lost and alone, and the thought of a helping hand, even a spectral one, was strangely appealing.

"I...I don't know," Lily said, her voice barely a whisper. "This is...this is a lot to take in."

"Take your time," Mr. Henderson said, his voice soothing. "I understand. It's not every day you meet a ghost who wants to give you business advice."

Lily couldn't help but smile at that, a flicker of humor breaking through her apprehension. "No, it's not," she agreed, shaking her head. "It's certainly not."

Mr. Henderson's smile broadened. "Perhaps we can have a cup of tea, get to know each other a little better?"

Lily hesitated, her mind a jumble of conflicting emotions. But something inside her, a spark of hope, a desperate yearning for a solution, pushed her forward.

"Alright," she said, her voice a little stronger now. "A cup of tea. But I'm not promising anything."

Mr. Henderson's smile remained, his eyes twinkling with amusement. "That's perfectly fine, Lily," he said. "That's perfectly fine."

And so, in the dusty, sun-drenched living room of her childhood home, Lily found herself sitting across from a friendly ghost, sharing a cup of tea and a conversation that would change her life forever. The road to success had always seemed long and winding, but with Mr. Henderson by her side, even if he was a little bit spectral, it felt like maybe, just maybe, she was finally on the right path.

Lily's food truck business was booming. Her once-empty parking lots were now filled with eager customers, the aroma of her signature dishes wafting through the air, drawing them in like moths to a flame. The local newspapers had featured her, raving about her innovative dishes and her unique, whimsical approach to cooking. She'd even won a prestigious award at a regional food truck competition, a testament to her talent and the power of Mr. Henderson's guidance.

But with success came a new set of challenges. The pressure to maintain her momentum, the constant demands of running a thriving business, the ever-present fear of losing her edge – these anxieties gnawed at her, leaving her feeling exhausted and overwhelmed.

One evening, after a particularly hectic day, Lily found herself back in the familiar, dusty living room of her childhood home. Mr. Henderson, his spectral form shimmering in the soft glow of the lamplight, sat beside her, a cup of tea warming his translucent hand.

"You're working too hard, Lily," Mr. Henderson said, his voice a gentle reprimand. "Remember what I told you about balance? You need to take care of yourself, to make time for the things that matter."

Lily sighed, her head resting in her hand. "I know, Mr. Henderson," she said, her voice weary. "But it's all so exciting, so overwhelming. I'm afraid of letting it all slip away."

Mr. Henderson smiled, his eyes twinkling with understanding. "You're not letting anything slip away, Lily," he reassured her. "You've built something special, something that will last. But you need to remember that success is a journey, not a destination. Enjoy the ride."

Lily looked at him, her heart heavy. "I don't know, Mr. Henderson," she said. "I feel like I'm losing myself in all this. I'm so focused on the food truck, on the business, that I've forgotten who I am."

Mr. Henderson placed a gentle hand on her shoulder, his spectral touch sending a shiver down her spine. "You're never going to lose yourself, Lily," he said, his voice a soft murmur. "Your passion for food, your creativity, your kindness – these are all part of who you are. Don't let the business overshadow what truly makes you special."

Lily's eyes welled up, a wave of emotion washing over her. She felt a deep sense of gratitude for Mr. Henderson, for his unwavering support, his gentle guidance, his ability to see through the noise and remind her of what truly mattered.

"What am I going to do, Mr. Henderson?" she asked, tears streaming down her cheeks. "I feel so lost."

Mr. Henderson smiled, his spectral form seeming to glow with a soft, warm light. "You're not lost, Lily," he said, his voice a whisper of reassurance. "You're exactly where you're supposed to be. Just remember to breathe, to take a step back, to appreciate the journey. And never, ever, forget who you are."

Lily sat in silence, her eyes fixed on the flickering flames in the fireplace. Mr. Henderson's words echoed in her mind, settling like a soothing balm on her troubled soul. She had been so focused on building her business, on chasing the elusive dream of success, that she had forgotten to take care of herself.

As the fire crackled and popped, a wave of exhaustion washed over her. The last few months had been a whirlwind of activity, a constant blur of cooking, serving, and strategizing. She had pushed herself to the limit, driven by a burning desire to prove herself, to make her grandmother proud. But in the process, she had lost sight of what truly mattered.

Lily stood up, stretching her stiff muscles. She needed to get out, away from the house, away from the weight of her anxieties. The desert night air, with its cool breeze and starry sky, might help to clear her head.

As she stepped out onto the porch, the silence of the night enveloped her. The only sounds were the distant howling of coyotes and the chirping of crickets. She took a deep breath, inhaling the scent of desert wildflowers and the cool, dry air. It was a reminder of the simple beauty that surrounded her, a beauty she had been too busy to notice.

Lily walked down the steps, her bare feet sinking into the soft sand. She began to walk, her footsteps echoing on the quiet street, the stars above twinkling like a million tiny diamonds. She walked until she reached the edge of town, where the desert stretched out before her, a vast, endless expanse of sand and sky.

She sat down on a weathered rock, her back against a gnarled mesquite tree. The stars twinkled above, casting a soft, ethereal glow on the desert landscape. Lily closed her eyes, trying to quiet the racing thoughts in her mind. She took another deep breath,

allowing the cool night air to fill her lungs.

As she sat there, contemplating her life, she realized that Mr. Henderson was right. She had been so focused on the destination that she had forgotten to enjoy the journey. She had let the pressure of success cloud her judgment, making her forget the things that truly mattered.

Lily opened her eyes, a new sense of clarity flooding through her. She didn't need to sacrifice her well-being for the sake of her business. She could have both – a thriving food truck and a happy, fulfilling life.

The desert night, with its vastness and silence, had given her the space she needed to reflect, to reconnect with herself. She had found a new perspective, a renewed sense of purpose. She would continue to work hard, to chase her dreams, but she would do so with a newfound balance, a commitment to taking care of herself.

As the first rays of dawn began to paint the sky with shades of pink and orange, Lily stood up and started to walk back towards town. She had a lot of work to do, but she felt a sense of excitement, a renewed energy. She had learned a valuable lesson from Mr. Henderson, a lesson that would guide her on her path to success.

The following morning, Lily woke up with a renewed sense of purpose. The sun streamed through the dusty window, painting the room in a warm, golden light. The weight of anxiety that had been pressing down on her seemed to have lifted, replaced by a feeling of calm clarity. She felt lighter, more grounded, as if the desert night had washed away the anxieties that had been clouding her mind.

She made herself a cup of strong coffee, the aroma filling the kitchen with a comforting scent. As she sipped her coffee, she thought about Mr. Henderson's words. He had reminded her of the importance of balance, of taking care of herself, of remembering who she truly was.

Lily took a deep breath, feeling a surge of determination. She would continue to chase her dream of a successful food truck, but she would do so with a newfound awareness of her own well-being. She wouldn't let the pressure of success overshadow her own happiness.

She decided to start by making some changes to her daily routine. She would take more breaks, schedule time for herself, and make sure to get enough sleep. She would also make an effort to connect with friends and family, to nurture the relationships that brought joy into her life.

Lily also decided to incorporate Mr. Henderson's advice into her business strategy. She realized that his insights, while sometimes outdated, were rooted in a deep understanding of food and a genuine love for culinary excellence. She would continue to experiment with new dishes, but she would also pay homage to the traditional recipes and techniques that Mr. Henderson had shared with her.

Lily spent the next few weeks putting her new plan into action. She hired a part-time assistant to help her with the workload, allowing her to take more breaks and spend more time with friends and family. She also started incorporating some of Mr. Henderson's recipes into her menu, adding a touch of old-world charm to her modern dishes.

The changes she made had a positive impact on both her personal life and her business. She felt less stressed and more fulfilled, and her customers seemed to appreciate the new additions to her menu.

One afternoon, as Lily was preparing a batch of Mr. Henderson's signature apple pie, she felt a familiar presence beside her. She glanced over, her heart skipping a beat. Mr. Henderson stood there, his translucent form shimmering in the afternoon light. He smiled, his eyes twinkling with amusement.

"You've been doing well, Lily," he said, his voice a gentle murmur. "I'm proud of you."

Lily smiled back, a wave of warmth washing over her. "Thank you, Mr. Henderson," she said, her voice filled with gratitude. "I couldn't have done it without your guidance."

Mr. Henderson chuckled softly, his ghostly form seeming to glow with a soft, warm light. "You've found your own way, Lily," he said. "You've taken my advice and made it your own. You've created something special."

Lily nodded, her heart filled with a sense of accomplishment. She had learned so much from Mr. Henderson, not just about cooking and business, but about life itself. She had learned the importance of balance, of taking care of herself, of following her own path.

As the days turned into weeks and the weeks into months, Lily's food truck continued to thrive. She expanded her operation, hiring more staff and opening a second location. She became a respected figure in the local culinary scene, known for her innovative dishes and her commitment to quality.

But even as she achieved her dreams, Lily never forgot Mr. Henderson. She kept his recipes close at hand, and she often shared stories of him with her staff and customers. She knew that he was always with her, a ghostly mentor and friend, guiding her on her journey.

One evening, as Lily was closing up her food truck for the night, she noticed a faint shimmer in the air. She looked up, her heart filled with a mixture of excitement and sadness. Mr. Henderson stood there, his ghostly form shimmering with a gentle light.

"It's time for me to go, Lily," he said, his voice a whisper. "But I'll always be with you."

Lily's eyes welled up with tears. "Thank you, Mr. Henderson," she whispered. "Thank you for everything."

Mr. Henderson smiled, his eyes filled with a warmth that transcended the boundaries of life and death. "You've made me proud, Lily," he said. "Now go out there and make your own magic."

And with that, Mr. Henderson faded away, leaving Lily with a sense of peace and gratitude. She knew that she would never forget him, that his presence would always be with her, a reminder of the power of connection, of the importance of following your dreams, and of the magic that can happen when you open your heart to the unexpected.

**Example Prompt**

Write a science fiction story about someone who is a time traveler and has dedicated everything in their life towards a goal, and now wonders if it was worth it. The story should be between 850 and 900 words. The story should begin with the main character waking up on a frozen tundra. He looks for shelter from the cold. He sees a dead wooly mammoth and realizes he traveled back to the ice age. The character should find shelter, and a predator is outside his shelter at night. The ending should not be happy.

**Human-written Story**

The cold wind pierced Robert's exposed skin like hundreds of bee stings. Sprawled on his stomach on the solid, frozen ground, he felt chilled to his core. The light from the sun reflected off the endless stretch of ice. He struggled to open his eyes in the glare.

Robert sat up and tried to regain his bearings. Although he was still foggy from the leap, he knew he must hasten to find shelter from the relentless cold.

"How could I be so ill prepared?" he berated himself aloud. The most he had ever traveled in time was forward seven months. He couldn't control the date and time of his destination, but surely he had not time-jumped beyond civilization. Robert stood up and steadied himself on the ice. He looked around at the vast white wilderness and shook his head. This was the moment. The moment he acted out in his backyard as a child. The moment he had given up sleep to study entanglement and wormholes for. The moment for which he had sacrificed everything. He always imagined more pomp and circumstance and less uncertainty.

Robert tightened the hood of his sweatshirt over his head and pulled his hands into his sleeves. With no buildings in sight, he decided his best bet was to walk along the nearby riverbed to find a crevice or overhang that could provide shelter. The ground was hard and slick with a light dusting of snow.

Sediment and rocks frozen on the surface helped provide some traction. A large dark mass appeared in the distance. As he neared the enormous object, the stench of rotten meat with the slightest note of sweetness grew stronger.

"No, no. It can't be." Robert audibly gasped.

Before him lay the ravaged carcass of a young wooly mammoth. Thoughts began to race and Robert grew dizzy. He fell to his knees before the massive tusks and began to dry heave. A combination of the putrid smell and the realization that he had actually traveled twenty thousand years into the past overwhelmed his mind and body. What had he gotten himself into?

Survival kicked in. He felt in his pocket for the hunting knife his grandfather gave him when he was a child. It had only ever been used to cut string or open packages. It had never been used on an actual animal. As rancid as the beast was, the fur would provide some protection from the biting cold. The skin was already a bit loose, and he cut through the ligaments to remove the pelt. He wrapped one piece around and felt immediate relief. He took another large piece to provide protection later.

Time travel took a huge toll on the body as it required a massive amount of energy. Pure adrenaline pushed Robert at this point. He must find shelter soon so that he could set up camp before sunset. There appeared to be a crevice in the rock beneath an overhang in the river bed. It was small but gave Robert enough room to stretch out. Not that he wanted to. All he wanted to do now was curl up under the pelt. He used large rocks to hold the second hide in place as a curtain in front of the opening.

He ventured only a few feet from his camp to gather branches from a fallen tree. A fire was critical, not only for warmth. It would hopefully keep away whatever had attacked that mammoth. He piled some wood near the crevice opening. He arranged some small sticks for tinder. Though he never got his fire making badge as a boy scout, he did use his glasses to burn ants when he was a child. Always playing God. What a fine mess that landed him in now.

He shivered from the cold but held his hand as steady as he could. The sun focused a beam of light on the tinder. A small trail of smoke appeared, and he blew on the glowing embers. Flames ignited, and Robert sobbed. He understood the exaltation the first human to discover fire must have felt.

The adrenaline was wearing off and reality began to set in. He sat in the shelter, hunched over with his arms wrapped around his legs. He stroked the wooly mammoth fur. When he had dreamed of time travel, he imagined watching the pyramids being built or leaping forward to see flying cars. Exhaustion finally overtook Robert, and he drifted off to sleep.

The fire had begun to smolder. Roberts' eyes popped open. Disoriented, he sat up, desperately searching for clues. Although he was completely surrounded by the blackness of night, the chill that ran through him reminded him of his reality. Outside the shelter, he heard a clicking on the ice. It stopped for a minute. Then it happened again.

A little louder and a little closer. Another pause. More tapping. The curtain moved and a sliver of moonlight shone through the opening. The light bounced off a long sharp fang, and Robert felt a hot breath on his arm. The creature let out a low growl. Robert closed his eyes tightly and prepared for the final leap into the unknown.

**System output:** E2E$_{ZS}$

The crunch of ice underfoot was the first thing I noticed. Then the biting wind, whipping across the frozen wasteland. I stumbled, the unfamiliar cold searing through my thin clothing. My breath formed white plumes in the air, instantly freezing on my face. I was on a frozen tundra, a vast, white expanse stretching to the horizon.

It had been a long journey, a journey spanning millennia. My life, consumed by this singular goal, had finally brought me here. The Ice Age.

I scanned the landscape, searching for any sign of shelter. A lone, skeletal tree stood stark against the blinding white. Then, a horrifying sight. A colossal carcass, its massive tusks jutting out from the snow, lay frozen in time. A wooly mammoth, its once-mighty form now a testament to the cruel indifference of nature.

I hurried towards it, the sight of the mammoth offering a glimmer of hope. Perhaps I could find some kind of shelter within its decaying remains. As I approached, the putrid stench hit me, a noxious mix of decay and death. I gagged, but pressed on. I had to find shelter before the sun dipped below the horizon and the cold truly bit.

Using my time-traveling device, I cut through the mammoth's tough hide, revealing a cavernous space within. The stench was overpowering, but it was better than freezing to death. I huddled inside, the decaying flesh a grim reminder of my isolation.

I felt a surge of exhaustion, a weariness that had built up over countless years of preparation, of relentless pursuit of this singular goal. Was it worth it? This desolate landscape, this chilling silence, this overwhelming sense of loneliness?

As darkness crawled across the sky, I huddled deeper into the carcass, the cold seeping through my clothes, a constant reminder of my vulnerability. The wind howled outside, a symphony of icy fury.

A guttural growl ripped through the silence. My heart leaped into my throat. A shadow flickered at the entrance of the makeshift shelter. My eyes, adjusting to the darkness, made out the silhouette of a massive beast, its amber eyes glowing in the moonlight. A saber-toothed tiger, its fangs glinting with deadly intent.

Panic surged through me. I had no weapons, no tools, no way to defend myself. I was trapped, a helpless prey in the jaws of an ancient predator.

I tried to back away, but the space was too small. The tiger, sensing my fear, pressed closer, its hot breath washing over me. It lowered its head, its fangs bared in a silent, deadly threat.

The smell of death, both from the mammoth and the tiger, filled the air. I closed my eyes, bracing for the inevitable. This was the end. The culmination of my life's work, the grand goal I had pursued with unwavering determination, had led to this - a gruesome death in the jaws of a prehistoric beast.

The tiger's growl, a low, guttural rumble, echoed in the cavern. I waited, frozen in fear, for the final, fatal blow. But it never came.

Instead, a different sound, a high-pitched whine, pierced the night. The tiger, startled, turned its head towards the sound. My heart pounded in my chest, a desperate hope flickering within me.

But then, silence. The tiger remained motionless, its amber eyes fixed on the darkness outside. The whine had stopped.

I held my breath, waiting for the attack to resume. But the silence held. Minutes stretched into an eternity, the tension in the air palpable.

Then, a faint, rhythmic thud, growing louder with each beat. The tiger's head turned, its eyes widening in fear.

The thudding grew into a thunderous roar. Something massive, something much larger than the tiger, was approaching.

I couldn't see what it was, but the tiger's fear was contagious. It backed away, its growls now a whimper of fear, its tail tucked between its legs.

The earth shook as the creature drew closer, its heavy footsteps echoing in the night. The tiger, sensing its own insignificance in the face of this new threat, turned and fled, disappearing into the darkness.

I remained huddled in the carcass, a silent witness to the unfolding drama. I had escaped the immediate threat, but a new, even more terrifying one had arrived. I was alone, trapped in a hostile world, with no idea what awaited me in the darkness.

The chilling roar of the unknown creature echoed through the frozen wasteland, a sound that reverberated through my bones, a sound that told me my journey was far from over. And the question, "Was it worth it?" echoed even louder, a haunting reminder of the sacrifices I had made, the price I had paid for this moment. And the answer, a cold, stark truth, whispered back: "No."

**System output: AGENTS' ROOM$_{ZS}$**

The world was a canvas of white, stretching out endlessly in every direction. Elias, his breath forming frosty plumes in the frigid air, stumbled forward, his boots sinking into the powdery snow. The wind, a relentless beast, whipped at his face, stealing the warmth from his exposed skin. He was alone, utterly and completely alone, on a frozen tundra that seemed to stretch back to the very dawn of time.

He had arrived. His journey, a life dedicated to a singular goal, had culminated in this moment. He had traveled back in time, escaping the sterile confines of his lab and the relentless hum of his time machine, to stand on the frozen plains of the Ice Age. It was a dream realized, a goal he had pursued with an almost religious fervor, sacrificing everything else in his life to achieve it.

But now, the reality of the situation crashed down upon him with the force of a glacial avalanche. The stark beauty of the landscape, the vastness of the frozen world, was quickly eclipsed by the crushing weight of his isolation. He was a lone figure, a speck of humanity lost in a world of ice and snow. His heart pounded against his ribs, a frantic drumbeat in the silence of the tundra.

He had to find shelter. The cold was relentless, biting at his exposed skin, sapping his strength. He trudged forward, his eyes scanning the horizon for any sign of respite, any hint of protection from the elements. The landscape was unforgiving, a vast expanse of white punctuated only by the occasional jagged rock protruding from the ice sheet.

Then, he saw it. A shallow cave, formed by a massive sheet of ice that had buckled and collapsed, offering a meager haven from the wind and the biting cold. He stumbled towards it, his body shivering with exhaustion and the chilling cold. He was desperate for warmth, for a moment of respite from the relentless assault of the elements.

As he entered the cave, the wind's icy fingers no longer reached him. The air inside was stale, heavy with the scent of frozen earth, but it was a welcome change from the biting wind that had been tearing at his face. The ice walls dripped and groaned, a constant reminder of the power of nature, the relentless force that ruled this frozen world.

He sank to the floor, his back against the cold, damp ice. He was exhausted, his body aching with the cold. He closed his eyes, trying to shut out the harsh realities of his surroundings. For a moment, he allowed himself to savor the victory, the culmination of his life's work. But the feeling was fleeting, quickly replaced by a growing sense of unease.

The isolation was suffocating. He was completely alone, cut off from the world he knew, the world he had left behind. He had sacrificed everything for this moment, for this journey, and now, he was starting to question if it was truly worth it. The beauty he had sought, the world he had dreamed of experiencing, was now a source of dread.

He opened his eyes, his gaze falling on a sight that sent a shiver down his spine. A woolly mammoth, its massive body frozen solid, lay sprawled in the snow just outside the entrance to his cave. Its tusks were long and curved, its eyes wide and vacant, staring into the endless expanse of the frozen tundra. It was a stark reminder of the harsh reality of this world, a world where life was a constant struggle for survival.

The mammoth's death was a chilling portent, a grim warning of the dangers that lurked in this unforgiving landscape. He was not alone. He was prey. And as the darkness settled over the frozen tundra, he could feel the presence of something else lurking in the shadows, a predator, watching, waiting.

The predator's presence was a constant in the back of his mind, a low hum of unease that vibrated through his bones. He couldn't shake the feeling of being watched, of being stalked. Every creak of the ice, every rustle of snow, sent a jolt of fear through him. He tried to rationalize it, to tell himself it was just his imagination, the product of his exhaustion and the isolation gnawing at his sanity. But the primal instinct for survival, dormant for so long in his sheltered life, now roared to life, demanding attention.

He spent the next day huddled in his cave, the silence broken only by the drip-drip-drip of melting ice and the distant howl of the wind. He tried to focus on the beauty of his surroundings, on the vastness of the frozen landscape, on the intricate patterns etched into the ice by the relentless wind. But the beauty was tinged with fear, a constant reminder of the power of nature, the untamed wilderness he had sought out.

He tried to read, to escape into the pages of a worn copy of Darwin's "The Origin of Species," a book he had carried with him, a symbol of the knowledge he had sought. But the words blurred, the sentences lost their meaning, and the pages seemed to mock him with their irrelevant tales of a world he could no longer comprehend.

He tried to sleep, to escape the reality of his situation in the embrace of unconsciousness. But sleep eluded him. He lay there, his eyes wide open, listening to the silence, the silence that was broken only by the occasional snap of a twig or the rustle of snow. He imagined the predator circling the cave, its keen eyes scanning the entrance, its powerful muscles coiled, ready to strike.

The fear was a physical sensation, a tightening in his chest, a clammy sweat on his skin. He was trapped, a lone figure in a world that had no room for him. He was a visitor, an intruder in a world that had no place for his kind.

As darkness fell, the fear intensified. The cold seemed to bite deeper, the wind to howl louder, the silence to press in on him, suffocating him. He huddled deeper into his cave, his back pressed against the ice, his hands clutching the tattered blanket he had brought with him. He was a small, insignificant creature, a speck of dust in the vastness of the frozen tundra.

He thought of his life, of the sacrifices he had made, of the years he had spent in pursuit of this moment. He thought of his family, of the friends he had abandoned, of the life he had left behind. He thought of the empty promises he had made to himself, the dreams he had chased, the goals he had achieved.

And now, he was here, alone, facing his own mortality. He was the product of his own ambition, a creature of his own making, and he was starting to realize that the price he had paid might have been too high.

The first crack in the ice came as a whisper, a barely audible groan that sent a tremor through the cave. Elias's heart leaped into his throat, his breath catching in his chest. He froze, every muscle in his body tense, listening. The silence stretched, a taut thread that

threatened to snap. Then, a second crack, louder this time, a distinct splintering sound that echoed through the frozen chamber.

He scrambled back, his eyes searching the ice wall for the source of the sound. A dark shape, a shadow against the pale light filtering through the entrance, began to take form. It was the wolf, its massive head wedged against the ice, its powerful jaws working at the frozen barrier. The animal's eyes, two burning orbs of yellow fire, were fixed on him, reflecting the meager light of the moon that had begun to peek through the clouds.

A low growl rumbled from the predator's throat, a sound that vibrated through the cave, a sound that spoke of primal hunger, of relentless instinct. Elias felt a wave of nausea rise within him, the taste of fear bitter on his tongue. He was trapped, a small, insignificant creature facing a predator that embodied the raw power of nature.

He was alone, completely alone, with no hope of escape, no chance of rescue. The years of sacrifice, the relentless pursuit of his goal, had brought him to this moment, a moment of absolute and utter despair. He had traded his life for this, for this frozen wasteland, for this terrifying confrontation with his own mortality.

His gaze fell on the mammoth's frozen body, its lifeless eyes staring into the vastness of the tundra. It was a mirror, reflecting his own fate, a testament to the harsh realities of this world. He had sought out this beauty, this unforgiving landscape, and now, he was about to become a part of it, a frozen relic in a world that had no place for him.

His fingers tightened around the tattered blanket, his last remaining comfort, his last connection to the world he had left behind. He was a prisoner in his own creation, trapped in a cage of his own making. He had sacrificed everything for this moment, and now, he was faced with the ultimate consequence of his ambition, a confrontation with the predator that lurked within himself.

The wolf's relentless assault on the ice wall was a symphony of destruction, each crack and groan a hammer blow to Elias's already shattered spirit. He pressed himself further into the back of the cave, his heart a frantic drum against his ribs. He was a cornered animal, facing a predator that seemed to embody the very essence of the Ice Age.

His gaze swept across the cave, landing on the mammoth's frozen form. It lay there, a silent testament to the brutal reality of this world, a world where life was a constant struggle for survival. The mammoth's fate was a chilling reflection of his own, a stark reminder that he was not immune to the laws of nature.

He tried to summon the courage to fight back, to defend himself against the relentless assault, but the fear was a heavy weight on his chest, suffocating his will. He was trapped, a small, insignificant creature facing the raw power of nature.

The wolf's growls intensified, a chorus of primal hunger that echoed through the cave. Each sound was a blow to his spirit, a reminder of the futility of his struggle. He closed his eyes, trying to shut out the sounds, the smells, the overwhelming sense of dread that permeated the air.

He thought of his life, of the years he had spent in pursuit of this moment, of the sacrifices he had made. He thought of the dreams he had chased, the goals he had achieved, and the emptiness that now filled his soul. He had traded his life for this, for this moment of terror, for this confrontation with the predator that lurked within himself.

The wolf's teeth ripped through the ice, sending a shower of shards flying. Elias flinched, his body trembling with fear. He could feel the cold air on his face, the smell of the predator's breath. He was on the brink, a precipice of despair from which there was no escape.

He wished he could go back, to undo the years of obsession, to reclaim the life he had abandoned. He wished he could find solace in the familiar comforts of his own time, in the warmth of human connection, in the simple joys of a life lived in balance. But those wishes were just echoes now, fleeting whispers in the deafening roar of his own despair.

The wolf's head finally broke through the ice, a jagged maw of teeth and fury emerging into the meager light of the cave. Elias, his body frozen by fear, could only stare as the creature's eyes, two burning embers of predatory hunger, locked onto his. The air within the cave seemed to crackle with anticipation, a palpable tension that hung heavy in the frigid air.

A final, guttural growl ripped from the wolf's throat, a sound that seemed to echo the shattering of Elias's own spirit. He knew, in that moment of terrifying clarity, that there was no escape. His journey, his life's obsession, had led him to this: a final, brutal confrontation with the harsh realities of the world he had sought to witness.

The wolf lunged, a blur of muscle and teeth, its powerful jaws snapping shut just inches from Elias's face. He screamed, a primal sound of terror that echoed through the cave, a desperate plea for a mercy that would never come. The wolf's teeth sank into his arm, a searing pain that ripped through him, a reminder that his journey had been nothing but a fool's errand.

He felt the wolf's weight on him, the creature's hot breath against his skin, the stench of raw, predatory instinct. He struggled, but it was a futile effort, a desperate dance against the inevitable. He could feel the life draining from him, the warmth of his blood mingling with the cold of the ice.

In his fading vision, he saw the mammoth's frozen form, a chilling reminder of the finality of his own fate. He had sought out beauty, knowledge, a glimpse into the past, but all he had found was an unforgiving wilderness, a world of constant struggle and a predator's hunger.

His last thought, as darkness claimed him, was a whisper of regret, a silent admission that his journey had been a tragic mistake, a pursuit of knowledge that had cost him everything. The predator's victory was complete, its hunger satisfied, its instinct fulfilled. Elias, the obsessed time traveler, was just another frozen relic in the vast, unforgiving landscape of the Ice Age, a testament to the dangers of ambition and the emptiness of a life dedicated solely to a singular goal.

# B    SPECIALIZED AGENTS' PROMPT TEMPLATES

Each specialized LLM-based agent has a specific prompt template that formats the scratchpad into a suitable prompt for the LLM call.

## B.1    SCRATCHPAD

Concretely, the scratchpad is typically formatted as follows:

---

**[SCRATCHPAD] Format**

[Creative Writing Task] <the original writing prompt>

[Central Conflict] <the output of the conflict agent>

[Character Descriptions] <the output of the character agent>

[Setting] <the output of the setting agent>

[Key Plot Points] <the output of the plot agent>

[Exposition] <the output of the exposition agent>

[Rising Action] <the output of the rising action agent>

[Climax] <the output of the climax agent>

[Falling Action] <the output of the falling action agent>

[Resolution] <the output of the resolution agent>

---

The number and order of items in the scratchpad is of course a function of which agents have been called so far.

## B.2    PLANNING AGENTS

---

**[CONFLICT] Agent Prompt**

Given <identifiers found in the scratchpad>, describe the central conflict in detail (more than 5 sentences). The description should answer the following questions:

- ⋆ What's the protagonist's main goal in this story?
- ⋆ Why do they want it?
- ⋆ What's stopping them from achieving it?

<scratchpad>

---

**[CHARACTER] Agent Prompt**

Given <identifiers found in the scratchpad>, describe the characters in detailed bullet points (more than 5 sentences for each character). The description should answer the following questions:

- ⋆ What do the characters sound like? Are they talkative or quiet? What kind of slang do they use? What is their sense of humor like?
- ⋆ What do they look like? Do they have any defining gestures? What's the first thing people notice about them?
- ⋆ What are their motivations and internal characteristics? What are their flaws? What are their values? What are they afraid of? How will they change and grow over the course of this story?

---

<scratchpad>

**[SETTING] Agent Prompt**

Given <identifiers found in the scratchpad>, describe the setting in detail (more than 5 sentences). The description should answer the following questions:

    ★ Where does the story take place? Is it set in a fictional world, or is it simply set in someone's backyard?

    ★ When does the story take place? What decade is it set in? How much time elapses over the course of the story?

<scratchpad>

**[PLOT] Agent Prompt**

Given <identifiers found in the scratchpad>, describe the key plot points in detailed bullet points.

<scratchpad>

The <identifiers found in the scratchpad> are extracted from the scratchpad and formatted to fit the prompt. For instance, for a scratchpad that contains the original prompt, the [CONFLICT] and [CHARACTER] agents' contributions, the resulting <identifiers found in the scratchpad> would be: "a Creative Writing Task, the Central Conflict, and the Character Descriptions".

## B.3 WRITING AGENTS

**[<SECTION>] Agent Prompt**

Given <identifiers found in the scratchpad>, continue the story by writing the <section> part.

<If previous sections have been written, include the following in the prompt:>

Begin your portion of the story in a way that naturally flows from the previous ending. Match the writing style, vocabulary, and overall mood of the existing text. Do not re-explain details or events that have already been described.

<If this is not the meant to be the last section, include the following in the prompt:>

Focus only on the <section> part of the story. Do not write about the following parts of the story. Do not end the story.

<scratchpad>

In these writing agents' prompt templates:

- <section> is one of "Exposition", "Rising Action", "Climax", "Falling Action", or "Resolution",
- <identifiers found in the scratchpad> are extracted from the scratchpad and formatted to fit the prompt. For these writing agents they are formatted as follows: "a Creative Writing Task, the Content Plan (Central Conflict, Character Descriptions, Setting, Key Plot Points), and the Previous Parts of the Story (Exposition, Rising Action, Climax)", where the enumerated elements correspond to what is in the scratchpad.

In the specific case of the AGENTS' ROOM [PLANNING] variant, with only the planning agents, we still need a single writing agent to finalize the story, which we denote as the [FINALIZER]. This [FINALIZER] agent uses the following prompt template:

> **[FINALIZER] Agent Prompt**
>
> Given <identifiers found in the scratchpad>, write a story using the information below.
>
> <scratchpad>

## C    PROMPT TEMPLATES FOR SYNTHETIC DATA GENERATION

For the planning agents, we use the same prompt templates as in Appendix B.2 to generate the synthetic training data, except in this case, we provide the gold standard data in the scratchpad. As a consequence, the scratchpad is formatted as follows:

> **[SCRATCHPAD] Format**
>
> [Creative Writing Task] <the original writing prompt>
>
> [User-Written Response] <the gold output>

The <identifiers found in the scratchpad> are formatted as "a Creative Writing Task and a User-Written Response".

For the writing agents, we use the following prompt template to split to gold standard stories into distinct sections:

> **[WRITING] Synthetic Data Generation**
>
> Split the following story into sections:
>
> * [Exposition]: The exposition gives the reader the background info they need to jump right into the story's world. This is often found towards the beginning of the story.
> * [Rising Action]: The rising action is the moments in the story that lead up to the climax — choices the main characters have made and the events happening that are at odds with the characters' goals. This is where the story builds and the reader begins to invest in the characters.
> * [Climax]: The climax is the primary turning point and what the story has been building towards.
> * [Falling Action]: The falling action is the period of time in a story that follows the climax and leads to the resolution. It can be used to clarify the events of the climax, ease any built-up tension, or wrap up loose ends.
> * [Resolution]: This is the end of the story. It answers the remaining unanswered questions in the plot. The resolution is also the time to show the next step in the characters' lives.
>
> For each section, give the section header (marked as [Exposition], [Rising Action], [Climax], [Falling Action], and [Resolution]) followed by the first sentence of that section, copied exactly from the story.
>
> [User-Written Response] <the gold output>

## D  HUMAN EVALUATION INSTRUCTIONS

For this task, you will be presented with a writing prompt and two short stories corresponding to this prompt. Your task is to compare the quality of the two stories across several dimensions. This is a judgment task rather than an annotation task. As such, you should use your own judgment when you assign ratings, calibrated by the rubrics we provide.

This rating task consists of three steps: (1) Compare the quality of the two stories across four dimensions. (2) Rate which story you preferred. (3) (optional) Leave comment / feedback on the stories. In the following we provide detailed instructions for each step:

### D.1  RATE THE QUALITY OF THE STORY

Your task is to compare the quality of two stories along four different dimensions (plot, creativity, development, language use), as described in the Rubric table below.

While the dimensions may have overlap and work in interdependent ways, they are intended to capture distinct aspects of what makes a good story. Therefore, a story may score highly in one dimension and poorly in another. Furthermore, the features (marked as a, b, c) that make up a dimension may be thought of as cumulative. For example, a story may have strong characters but suffer from an underdeveloped setting.

The Rubric table is intended to help you calibrate your judgment so that you can roughly determine when a story is very good or even excellent along a particular dimension because it exhibits all of the features of that dimension. Conversely, if a story fails to exhibit most or all of the features of a dimension, then you may score the story as being poor or very poor along that dimension. The features are meant to be illustrative but not exhaustive; you may determine that a story should score poorly or well due to the absence or presence of additional features for a given dimension based on your judgment.

Another important thing to note about the features that make up the dimensions we're asking you to rate is that they describe conventions that may be followed or flouted; a story may contain intentional plot devices like non-linear timelines, discontinuity, and other stylistic choices to create effects. As with other features, these elements of a story should inform your judgment on their own merit (so that they only negatively impact your rating if they are ineffective or confusing and positively impact your rating if they are used well to make the story more interesting and unique).

We use a 3-point comparative rating scale for each of the dimensions. The rating scale can be thought of as described below:

| Rating | |
|---|---|
| A is better | Response A is better than Response B in that dimension. |
| About the same | Both responses are about the same in that dimension. |
| B is better | Response B is better than Response A in that dimension. |

The focus of this rubric is the quality of the writing, and not how well the stories follow the writing prompt. In particular, when rating with this rubric, we encourage you not to focus on the number of words mentioned in the writing prompts, but rather on the features described in the table below.

| Dimension | Features |
|---|---|
| Plot | a. The story has a recognizable structure, e.g. with a connected beginning, middle, and end.
b. The story exhibits events and turns that move the plot forward.
c. The story does not have logical or conceptual inconsistencies. Surprising or disruptive elements are intentional, e.g., they serve the story and do not feel jarring, odd, or out of place. |
| Creativity of Ideas, Themes, and Topics | a. Engaging characters, themes, and imagery. The ideas do not feel generic or bland.
b. Avoidance of overly cliched characters and storylines, unintentional tropes, and stereotypes. When used, tropes and cliches serve a purpose (e.g. comedic effect, twist on a common trope etc).
c. The story includes original elements that were not explicitly mentioned in the prompt. |
| Development | a. Characters and settings are introduced and contextualized with relevant details that allow the reader to understand their place in the story.
b. Appropriate levels of detail and complexity are provided to lend the story a feeling of realness and believability.

*Reminder: The features that make up a dimension may be thought of as cumulative. A story with a well-developed character, but in a lackluster setting (or vice-versa) would score lower in Development than a story that does well on both aspects.* |
| Language Use | a. The language used feels varied and rich: Variance of sentence structure, verbiage, and vocabulary.
b. The story exhibits rhetorical, linguistic and literary devices (e.g. ambiguity, alliteration, etc) to create interesting effects
c. The story avoids bland or repetitive phrases (unless used intentionally to create a narrative, thematic, or linguistic effect). |

We provided examples rated along these rubrics. While the examples include explanations, these are there as an aid, and you are not requested to provide explanations for your ratings.

### D.2 WHICH STORY DO YOU PREFER?

Do you find the story interesting, engaging, funny, or emotionally-rich? In addition to getting your judgments of the dimensions, we would also like to know whether you enjoyed reading the story. Similar to the dimensions, we will ask you to score which story you prefer:

- A is better

- About the same

- B is better

When rating, do not hesitate to be very critical.

### D.3 OPTIONAL: LEAVE COMMENTS OR FEEDBACK ON THE STORIES

Thank you for completing the ratings! If you have any additional comments or feedback you would like to provide on the story, feel free to add them in the "comments" section.

## E PROMPT TEMPLATE FOR THE LLM EVALUATOR

The following prompt template is used by the LLM to evaluate two system outputs *side-by-side* (we replace <story a> and <story b> with the two stories being evaluated):

[LLM EVALUATOR] Prompt Template

You will conduct a side-by-side evaluation. You will be given two system-generated stories. Your task is to compare the two stories and determine which one is better based on the following dimensions:

- *Plot:* The story should have a recognizable structure, e.g., with a connected beginning, middle, and end. The story should exhibit events and turns that move the plot forward. The story should not have logical or conceptual inconsistencies. Surprising or disruptive elements should be intentional, e.g., they serve the story and do not feel jarring, odd, or out of place.

- *Creativity:* There should be engaging characters, themes, and imagery. The ideas should not feel generic or bland. There should be avoidance of overly cliched characters and storylines, unintentional tropes, and stereotypes. When used, tropes and cliches should serve a purpose (e.g., comedic effect, twist on a common trope etc). The story should include original elements that were not explicitly mentioned in the prompt.

- *Development:* Characters and settings should be introduced and contextualized with relevant details that allow the reader to understand their place in the story. Appropriate levels of detail and complexity should be provided to lend the story a feeling of realness and believability.

- *Language Use:* The language used should feel varied and rich: Variance of sentence structure, verbiage, and vocabulary. The story should exhibit rhetorical, linguistic and literary devices (e.g., ambiguity, alliteration, etc) to create interesting effects. The story should avoid bland or repetitive phrases (unless used intentionally to create a narrative, thematic, or linguistic effect).

Provide a detailed assessment of the two stories in terms of these four dimensions. Conclude your assessment with scores for each dimension using the template below. Do not add any emphasis, such as bold and italics, on your assessment.
[Assessment Ending Template]

Based on my assessment, the better story for each dimension is:

Plot:           [A or B or Same]
Creativity:     [A or B or Same]
Development:    [A or B or Same]
Language Use:   [A or B or Same]
Overall:        [A or B or Same]

[Story A]
<story a>

[Story B]
<story b>

[Assessment]

Below we show example output from the LLM evaluator from which we extract preference scores. In this case, the evaluator preferred story A on all dimensions.

---

**[LLM EVALUATOR] Example Output**

Story A has a more defined plot structure with a clear beginning, rising action, climax, and resolution. The summoning of Horus, his reaction to the modern world, and his eventual acceptance create a compelling narrative arc. Story B, while touching upon interesting themes, lacks a clear climax and resolution. The story feels incomplete, leaving the reader with unanswered questions about Horus's fate and Ash's journey.

Story A exhibits more creativity in its portrayal of Horus and his interactions with Amelia. The falcon god's bewilderment, anger, and eventual acceptance are depicted with nuance and depth. The story also introduces original elements, such as the restoration of the canopic jar, which adds a layer of symbolism and reinforces the theme of reconciliation. Story B relies on more generic depictions of a god's reaction to the modern world, lacking the originality and depth seen in Story A.

Both stories provide adequate development of their main characters, Amelia and Ash, respectively. However, Story A excels in its development of Horus, providing a more nuanced and believable portrayal of a god grappling with the realities of a changed world. The story delves into his emotions, his struggles, and his eventual acceptance, making him a more compelling and relatable character. Story B's portrayal of Horus, while touching upon his disillusionment, lacks the same level of depth and complexity.

Story A utilizes a richer and more varied language, employing vivid imagery, sensory details, and a wider range of vocabulary. The story effectively uses rhetorical devices, such as repetition and contrast, to emphasize the differences between the ancient and modern worlds. Story B, while adequately written, lacks the same level of linguistic richness and sophistication.

Based on my assessment, the better story for each dimension is:

| | |
|---|---|
| Plot: | A |
| Creativity: | A |
| Development: | A |
| Language Use: | A |
| Overall: | A |

---

## F  ADDITIONAL RESULTS

While we consider human-based evaluation our primary means of evaluation, the LLM evaluator helps us assess overall system-level trends. We report pairwise win rate (proportion of examples on which our AGENTS' ROOM *plan + write* variant performed better than comparison systems according to our LLM-based evaluator. Table 3 complements Figure 3a.

| $AR_{ZS}$ plan + write vs | overall | plot | creativ. | develop. | language |
|---|---|---|---|---|---|
| $E2E_{ZS}$ plan | 74.55 | 63.64 | 75.47 | 75.93 | 81.13 |
| $E2E_{ZS}$ reflect | 67.27 | 63.64 | 67.92 | 68.52 | 69.23 |
| $E2E_{ZS}$ decompose | 89.09 | 80.00 | 87.04 | 90.91 | 90.91 |
| 2STAGE decompose | 66.67 | 59.26 | 64.15 | 66.67 | 67.92 |

Table 3: Proportion of times LLM-based evaluator preferred $AR_{ZS}$ to comparison system across overall, and across the dimensions of plot, creatitivy, development, and language use.

The majority of our experiments were conducted using a Gemini 1.5 Flash backbone model. This choice was dictated by the nature of the creative writing task which is challenging to accomplish with models that do not have a long enough context window and adequate writing quality. Most recent work on storytelling using a single model resorts to large, proprietary models such as GPT

(Yang et al., 2023; 2022), or Claude (Chakrabarty et al., 2024a). This is also the case for multi-agent systems targeting writing which seem to be exclusively relying on GPT-4 (Chen et al., 2024; Bai et al., 2024). Nevertheless, using Gemma2-9B-it (Riviere et al., 2024) as a backbone model we compare AGENTS' ROOM and E2E systems in the zero-shot setting, using the LLM-based evaluator.

| AGENTS' ROOM$_{ZS}$ plan+write vs. | overall | plot | creativ. | develop. | language |
|---|---|---|---|---|---|
| E2E$_{ZS}$ | 80.00 | 67.27 | 84.62 | 83.33 | 77.78 |

Table 4: Proportion of times LLM-based evaluator preferred AGENTS' ROOM$_{ZS}$ to E2E$_{ZS}$ overall, and across the dimensions of plot, creativity, development, and language use.

As can be seen in Table 4, even with the smaller scale Gemma2-9B-it model, AGENTS' ROOM greatly outperforms the end-to-end baseline across all dimensions of evaluation.

Finally, although we did not elicit feedback on individual story dimensions, we did ask participants to comment on the quality of the stories produced by our systems, and possibly on aspects of story quality our instructions did not cover (see Section D). We show some of this feedback below.

---

**Participant Feedback**

"The task was interesting, but over time, I found the language redundant. There seemed to be a go-to vocabulary list utilized in the majority of the stories, phrases used time and again, making the output somewhat predictable."

"It was interesting to see what kind of fictional narrative the model would generate. Most of the stories seemed to be written at a seventh grade level. The stories didn't stray too far from the input and for the most part were grammatically correct. There were at times, instances of repetitiveness, including entire paragraphs, that made me wonder what the model was doing."

"The stories showed some promise, but often fell into the same pitfalls of loops or sudden tone discordance. . . "

---

