# OpenReview forum: "Agents' Room:  Narrative Generation through Multi-step Collaboration"
_ICLR.cc/2025/Conference — ICLR 2025 Poster_

### Official Review · Reviewer_ctY6 · 2024-11-02

**Soundness:** 3
**Presentation:** 4
**Contribution:** 4
**Rating:** 8
**Confidence:** 5

**Summary:**

This paper introduces AGENTS' ROOM, a framework that breaks down complex writing tasks into subtasks handled by specific AI agents working collaboratively. The framework includes planning agents that develop story elements like plot and characters, and writing agents that generate specific narrative sections, all coordinated by a central orchestrator that manages information flow through a shared scratchpad. To demonstrate the framework's effectiveness, authors created TELL ME A STORY, a high-quality dataset of writing prompts and human-written stories developed through writing workshops, which they use to train and evaluate their system. They implement their framework using large language models as agents and explore both zero-shot and fine-tuned approaches, showing that AGENTS' ROOM generates stories preferred by human evaluators over baseline systems. Their evaluation framework assesses multiple dimensions of story quality, including plot, creativity, development, and language use, along with an LLM-based evaluator that correlates significantly with human judgments.

**Strengths:**

There is so much to love about this paper. Great Motivation. Great Design. Great Evaluation. Great Technical Depth. One of the fundamental bottlenecks of autoregressive language models is that they do not look at the bigger picture, hierarchical structure that often drives narratives. If you simply prompt an LLM to write a story it will generate something typically shorter than what you want and often misses out on crucial details because these models aren't fundamentally good at Narrative Planning (https://arxiv.org/abs/2402.01817).  So it makes total sense to break this into agents. It reminded me of Operating systems. Just like how multithreaded systems break down complex tasks into parallel threads, AGENTS' ROOM decomposes the writing task into specialized subtasks (like plot planning, character development, etc.) handled by different agents. The shared memory , orchestrator, State Management etc are all coming from OS and I find the design very neat for a totally different task. I very much appreciate detailed descriptions of agent prompts in Appendix that is very helpful for reproducibility. Its great that authors employ similar practices as that of WordCraft, Dramatron etc to get professional writers come of with fresh prompts and flash fiction stories that are not contained in Pre-training  (yet ? hard to stop from contamination in future).
I like the evaluation done by experts mostly preference evaluation that is very popular in Alignment, Behavioral Science literature. This adds further credibility. Nice ablations. Additional auto eval showing repetition through ngrams. Great to see fresh human written stories do better than models here. Last but not the least nice to include LLM evaluation given current trends for RLAIF (Bai et al) [ Though I would not trust them].

**Weaknesses:**

I don't think these are big weaknesses still

1) Since you are anyway having multi step collaboration and agents why skip Revising ? Flower and Hayes's Cognitive process model of writing includes Planning , Writing and Revising. I am sure expert writers revise when they write. In professional setting a Literary agent takes on that role suggesting both developmental and line level edits. This would add more credibility to the paper and worth discussing this in camera ready

2) Length as a confounder : I believe given that your stories have very different length, how this translates to eval? To control for this (Chakrabarty et al 2024) did iterative expansion till all stories had comparable length. My point is experts could dismiss shorter length AI generated stories as they are both Poor Quality + Brief. Of-course expert writing is infinitely better so experts can recognize that even if 50% shorter

3) Could be interesting to show the diversity of stories in your data. What are the genres ? Are there any recurring motifs or themes.WritingPrompt has a disproportionately high number of Alien stories. We don't want the case to be that.

**Questions:**

NA

---

> ### Author Response · Authors · 2024-11-19
> **We thank the reviewer for their great review and thoughtful feedback**
>
> We thank the reviewer for their great review and thoughtful feedback. We are delighted by the reviewer's enthusiasm for all the dimensions of our work, motivation, design, evaluation, technical depth! In the following, we elaborate on their comments:
>
> 1. *About revising:* We considered revising and editing agents but we first had to develop an evaluation framework for long-form narratives that was detailed and robust enough to capture subtle differences between narratives. Therefore, this current paper has a strong data and evaluation component. In future work, we would like to explore revising and editing in more depth. We will include some discussion around this in the camera-ready version.
>
> 2. *Length as a confounder:* Our evaluation rubrics (overall, plot, creativity, development, and language use) were specifically formulated to not be a function of the story length. In addition, the human raters were explicitly instructed to not use the length as a discriminating criteria.
>
> 3. *Diversity of stories in the dataset:* We will report a breakdown of the genres in our dataset. The majority of the stories belonged to the genres of science fiction and fantasy. We also included stories from the following genres: horror, drama, comedy, adventure, and folklore. We should also point out that stories often belonged to several genres (e.g., comedic horror or urban fantasy). We will add more discussion in the paper.

---

> > ### Author Response · Authors · 2024-11-20
> >
> > Following the reviewer’s suggestion about length as a confounder: We computed the proportion of pairwise comparisons for which our human raters preferred the longer story overall (excluding ties) and found it to be around 0.5058.

---

> > ### Comment · Reviewer_ctY6 · 2024-11-21
> > **Thank You**
> >
> > Amazing. As long as you discuss in Camera ready its great. It doesnt reduce the value of your work but makes it stronger

---

> > > ### Author Response · Authors · 2024-11-26
> > >
> > > We thank the reviewer again for their detailed review, the comments were very valuable. We will include these elements in the camera-ready version.

---

### Official Review · Reviewer_pcJk · 2024-11-02

**Soundness:** 2
**Presentation:** 3
**Contribution:** 3
**Rating:** 5
**Confidence:** 5

**Summary:**

This paper conceptualize long-form writing as a multi-agent collaboration problem. Leveraging narrative theories and existing multi-agent works in other domains, it decomposes the writing task into sub-tasks tackled by specialized agents (a few for planning and others for writing), which communicate via scratch pad and a central controller called orchestrator.

In addition, the authors contributed TELL ME A STORY, a high-quality dataset of prompts and longform stories collected through multiple rounds of writing workshops with human writers.

**Strengths:**

1) This paper proposes a novel multi-agent multi-step framework for long-form writing, an important task for content generation with LLM
2) It also contributes a dataset containing human generated prompts and stories, which I assume to be high-quality
3) The paper is well-written and easy to follow

**Weaknesses:**

I am most concerned with the soundness of the experiments that currently do not support the claims made. To name the most severe ones:

1) **Effectiveness of multi-agent collaboration**: One of the most fundamental motivating claims is that single-agent works, such as detailed prompts to guide the generation process (Yang et al., 2022; Xie et al., 2023), prompt chaining (Mirowski et al., 2023; Yang et al., 2022), and planning strategies (Yang et al., 2023; Lee et al., 2024) fail to generate high-quality stories. Additionally, this paper which I found interesting [1] also showed that prompting a single LLM to reason about turning points improves overall narrative construction such as reducing plot holes and generating better plot twists.

However, none of these methods are compared against in the experiments. I would expect to see at least 2~3 baselines from the above to verify if indeed the improvement is **because of** multi-agent collaboration. The authors just compared to an end-to-end approach, which is far from enough.

2) I am not fully convinced that "The LLM evaluator agrees with humans and itself". Based on figure 3, the left and right show very different trends for human and the best method (AR plan+write). Clearly, LLMs prefer its own output over human-written, whereas human evaluators prefer human-written ones. I doubt if the seemly high correlation is a result of taking the mix of extremely bad responses to boost the correlation. If we are already at a time where LLMs can generate moderately good stories, the LLM based evaluators seems highly unreliable.


[1] Are Large Language Models Capable of Generating Human-Level Narratives? EMNLP 2024

**Questions:**

1. Who are the participants of the writing workshop? What are their demographics? What are their backgrounds/expertise in writing and are they allowed to use writing assistant for writing/review/discussion? How motivated are they to produce the best writing pieces?
2. See weakness. I would consider raising my scores if Weakness 1 has been adequately addressed.

---

> ### Author Response · Authors · 2024-11-19
> **We thank the reviewer for their detailed comments and thoughtful feedback.**
>
> We thank the reviewer for their detailed comments and thoughtful feedback. We appreciate that the reviewer found our multi-agent system novel and considers long-form writing an important task for generation with LLMs. The reviewer also highlighted our human-written dataset contribution. In the following, we address their comments and provide answers to their questions.
>
> 1. *Effectiveness of multi-agent collaboration*
>
> In terms of baselines, we would like to highlight that one of the single-agent baselines presented in the paper is actually LoRA-tuned on high-quality data (human-written gold-standard training data), which is a strong baseline. We show that all the multi-agent systems that include writing agents outperform this LoRA-tuned baseline, even when using prompted agents only.
>
> In addition to these baselines, we will provide 4 additional baselines:
>
> - A. Two baselines using detailed prompting (following Yang et al., 2022; Xie et al., 2023):
>   - A1. The model is instructed to generate the central conflict, characters, setting, and plot according to detailed guidelines, before generating the story.
>   - A2.  The model is instructed to think about the central conflict, characters, setting, and plot according to detailed guidelines, before generating the story.
>
> For both A1. and A2., the detailed guidelines are the same as those provided in Appendix B.2.
>
> - B. Two baselines relying on planning strategies (following Yang et al., 2023; Lee et al., 2024):
>   - B1. The model is instructed to generate a plan before generating the story in one call.
>   - B2. The model is first instructed to generate a plan for the story, followed by a second call in which the model is instructed to generate the story based on the input prompt and the plan.
>
> We will update this thread shortly with the corresponding metrics.
>
> Some of the baselines mentioned by the reviewer can be seen as simpler variants of multi-agent systems, since they involve multiple agent calls with information passed from one agent to the next. As such, they are not purely “single”-agent setups. For instance, one of our ablations, the Agents’ Room zero-shot with only planning agents, corresponds to the prompt-chaining baseline mentioned by the reviewer. Our experiments show that this is the weakest of our multi-agent settings.
>
> As for the turning points (following [1] as suggested by the reviewer), the Agents’ Room framework is modular and allows for varying subtask decompositions. Therefore our proposed writing agents can be replaced with writing agents based on turning points. We actually experimented briefly with turning points, but our structure based on narrative theory (Freytag, 1896; Pavis, 1998) yielded better results.
>
> Overall, we would like to highlight that our proposed approach goes beyond these prompted methods, since we show that we can finetune and specialize the individual agents to improve the generated outputs, i.e. the different agents differ not only by the prompt, but they can be different specialized models.
>
> 2. *Regarding the use of LLM as an evaluator*
>
> Many previous studies [1] have highlighted the challenges associated with evaluating narratives automatically. Metrics based on lexical matching correlate poorly with human judgments ([2], [3]) and do not effectively measure story quality (e.g., is the story original and well-written with plausible characters). Although we consider human-based evaluation our primary means to assess system output, we use the LLM evaluator for system development. Our correlation analysis shows that the LLM can be used for system-level analysis rather than to rate individual stories. Indeed Figure 2 shows that the LLM obtains broadly similar system rankings compared to the humans (although as the reviewer points out, it is biased towards its own output). We will discuss this point further and also make reference to very recent work [4] corroborating our findings. Specifically, Chhun et al. perform a large scale study on the ability of LLMs to evaluate story quality and report results similar to ours: LLMs show very high self-consistency, while correlations with human ratings remain weak but LLMs display some improvement over non-LLM automatic measures.
>
> [1] https://arxiv.org/abs/2408.14622
>
> [2] https://aclanthology.org/2022.coling-1.509/
>
> [3] https://doi.org/10.1145/3613904.3642731
>
> [4] https://aclanthology.org/2024.tacl-1.62/
>
> **Answers to questions**
>
> 1. The participants of the writing workshop all have some type of advanced background in writing. Many have graduate degrees in English, some have professional writing experience, and many are former teachers.  They did not use a writing assistant for this task.
> 2. Cf our discussion about baselines above. We will follow up shortly with the additional baselines’ metrics.

---

> > ### Author Response · Authors · 2024-11-20
> > **We provide 4 additional experimental baselines**
> >
> > Following the reviewer’s suggestion, we now report metrics on **4 additional experimental baselines**:
> >
> > - A. **Two baselines using detailed prompting** (following Yang et al., 2022; Xie et al., 2023), which we denote as A1. and A2. (see detailed description in the previous comment).
> > - B. **Two baselines relying on planning strategies** (following Yang et al., 2023; Lee et al., 2024), which we denote as B1. and B2.  (see detailed description in the previous comment).
> >
> > While we consider human-based evaluation our primary means of evaluation, the LLM evaluator helps us assess overall system-level trends. We report pairwise win rate (proportion of examples on which our Agents’ Room (AR) framework performed better than the baseline) according to our LLM evaluator:
> >
> > |  | evaluation rubric |  |  |  |  |
> > | :---- | ----- | :---- | :---- | :---- | :---- |
> > | winrate | overall | plot | creativ. | develop. | language |
> > | AR vs A1. detailed prompt | 74.55 | 63.64 | 75.47 | 75.93 | 81.13 |
> > | AR vs A2. detailed prompt | 67.27 | 63.64 | 67.92 | 68.52 | 69.23 |
> > | AR vs B1. planning strategies | 89.09 | 80.00 | 87.04 | 90.91 | 90.91 |
> > | AR vs B2. planning strategies | 66.67 | 59.26 | 64.15 | 66.67 | 67.92 |
> >
> >
> > We observe that our Agents’ Room (AR) framework greatly outperforms all of these additional baselines across all dimensions of evaluation. We also compute our automated metrics on these new baselines, which we report below. As far as story length is concerned, we observe that these baselines are generally shorter than the stories produced by our Agents’ Room framework with writing agents.
> >
> > | model | \#words | \#para | article | pro | unique | intra | inter | overlap | rouge | bertscore |
> > | :---- | :---- | :---- | :---- | :---- | :---- | :---- | :---- | :---- | :---- | :---- |
> > | A1 | 1130 | 27.24 | 15.34 | 42.25 | 45.93 | 23.59 | 29.41 | .0027 | 20.58 | .8173 |
> > | A2 | 1126 | 28.62 | 13.79 | 40.96 | 45.85 | 23.68 | 23.95 | .0032 | 20.36 | .8152 |
> > | B1 | 965 | 21.25 | 21.36 | 39.62 | 45.49 | 31.98 | 44.10 | .0034 | 19.41 | .8067 |
> > | B2 | 1090 | 24.82 | 15.59 | 42.15 | 44.50 | 21.54 | 24.26 | .0031 | 20.35 | 0.816 |
> > | AR (ours) | 3034 | 58.65 | 15.97 | 41.43 | 35.05 | 44.73 | 43.25 | .0022 | 17.57 | .8123 |
> >
> > Caption for these automated metrics (same as in the paper): #words (average number of words per story), #para (average number of paragraphs per story), Article (percentage of sentences starting with an article), Pro (percentage of sentences starting with a pronoun), Unique (percentage of unique words), Intra (intra-story trigram repetition), Inter (inter-story trigram repetition), Overlap (proportion of trigrams overlapping with the prompt). We also report two reference-based metrics,  Rouge-L and BertScore.
> >
> > We believe these additional experiments address all the comments from the reviewer.

---

> > > ### Author Response · Authors · 2024-11-22
> > >
> > > Dear reviewer, as the discussion period is nearing its end, we would like to inquire if you have had the opportunity to review our responses to your comments. We value your feedback very much and we hope the additional experiments we provided addressed your comments.

---

> > > > ### Comment · Reviewer_pcJk · 2024-11-25
> > > >
> > > > Thanks the authors for their additional experimental results. I have raised my scores accordingly.

---

> > > > > ### Author Response · Authors · 2024-11-26
> > > > >
> > > > > We thank the reviewer for raising their scores. We greatly valued the detailed feedback and are glad the additional experimental results provided more baselines.

---

### Official Review · Reviewer_8uSS · 2024-11-03

**Soundness:** 3
**Presentation:** 4
**Contribution:** 3
**Rating:** 6
**Confidence:** 3

**Summary:**

This paper proposes a multi-agent framework for collaborative creative writing called Agent’s Room. In this framework, planning and writing agents pass along a scratchpad of outputs to collaboratively write a story. Planning agents first plan out a story’s main conflict, character/s, setting, and plot. Then, the system calls writing agents that specialize in a story’s exposition, rising action, climax, falling action, and resolution. These agents are fine-tuned on synthetic data generated by a larger language model, which decomposes human-written stories into relevant parts. Next, the paper evaluates generated stories by Agent’s Room, planning/writing ablated approaches, and an end-to-end baseline in a pairwise manner with human writers and an LLM-based evaluator. The authors find that though human-written stories still outperform LLM-based ones based on human judgements, Agent’s Room performs better than end-to-end story generation based on both human and LLM judgements.

**Strengths:**

This paper is well-written and thoughtful, and it’s evident that effort was put in to involve actual human writers in data collection and evaluation components. Generally I found this work to be pretty solid and empirically shows the utility and promise of currently trendy approaches in AI (e.g. collaborative agents, synthetic data generation, LLMs-as-judge) for creative writing.

**Weaknesses:**

This paper spends a considerable amount of space discussing and advocating for a “general” writing framework (e.g. lines 231-233 as well as the paragraph introducing the orchestrator), but it only provides empirical evidence of its utility for one task: fiction writing. So, it may be helpful to rework the paper’s framing so that its claims actually match with its findings. Discussion around the generalizability of the proposed approach could be left in the conclusion as future work. If the authors want to stick with proposing a general framework, then they need to run similar experiments with additional tasks.

Figure 1 displays little icons suggesting people, but the orchestrator, planning agents, and writing agents in this work are not people. It may be better to use a different icon. In addition, it may be useful to label agents with the roles you gave them in this current work; though the diagram may be intended to show a generalizable framework, its minimalism makes it somewhat uninformative.

There is also the dreaded “you could have experimented with more models” weakness that comes with any work that only uses one family of models (in this case, Gemini). I mostly note this because including an open model in the mix could improve the accessibility of this work (e.g. there is a practical difference between paying-per-token versus running open models on your own compute infrastructure). It can also increase the longevity of this work if its findings are explicitly shown to be non-dependent on a specific model, allowing more focus on the framework proposed. Alternatively, if other models are not strong enough to collaborate in the manner proposed by this paper, that in itself may be interesting to note.

Another suggestion is instead of spending space on Figure 2 showing three different prompts, it may be more interesting to show instead one prompt, the beginning of a story generated by an end-to-end system, and the beginning of one generated by your proposed system. That is, this entire paper discusses writing stories but doesn’t really ever show an example of a story or story excerpt. If finding space for this is an issue, I’m also not sure if Algorithm 1 really adds much to the paper and could be removed.

Finally, it may be useful to do some more reflection on why we would want models to write creatively on their own in the first place, and the (potentially negative) impacts this may have on human writers and the writing industry. This is, why would we want automatic writing systems that go beyond writing assistants? This isn’t to suggest that there are no possibly positive uses of automatic writing systems, but to encourage the authors to consider pros and cons of the basis of their work more thoroughly. Any technology that has the potential to replace rather than support humans surfaces risks which are not really discussed in this paper’s ethical considerations section. The authors do mention that components of their system could be interchanged with human writers to become a human-in-the-loop system, but don’t provide empirical evidence of this.

**Questions:**

How much experience with creative writing do the human writers have prior to participating in writing workshops?

It’s interesting that human judges prefer the system that involves writing agents alone over the one that has both planning and writing agents, while the LLM evaluator does not. Is there any intuitive/qualitative explanation as to why this may be the case? Did the human judges provide reasons for their judgements?

---

> ### Author Response · Authors · 2024-11-19
> **We thank the reviewer for their detailed comments and thoughtful feedback.**
>
> We thank the reviewer for their detailed comments and thoughtful feedback. We are glad to hear that the reviewer found our work thoughtful and solid and that they appreciated the involvement of actual human writers in the data collection and evaluation process. In the following, we address their comments and provide answers to their questions.
>
> - *Regarding the general writing framework:* We formalized definitions of all components since many of these terms, such as “agents”, tend to be overloaded. While we acknowledge that this paper demonstrates the utility of this framework only for fiction writing, we believe this use case is a strong one because of  its broadness and open-endedness. Adding further tasks, along with data, appropriate evaluations, etc. is beyond the scope of a single paper. We will move the discussion around generalization (such as lines 231-233) to the conclusion section.
>
> - *Figure 1:* We will pick different icons to represent the agents and we will add labels under the individual agents.
>
> - *Regarding experimenting with more models:* There are  some constraints on the choice of model backbone to actually achieve good performance on the creative writing task. Indeed, the model has to have both a long enough context window and high enough writing quality. Most recent work on storytelling using a single model resorts to large, proprietary models such as  GPT ([1], [2]) or CLAUDE [3]. This is also the case for multi-agent systems targeting writing which seem to be exclusively relying on GPT-4 ([4], [5]). Nevertheless, we are in the process of obtaining results with Gemma 2 9B to address the reviewer’s point.
>
> [1] https://aclanthology.org/2023.acl-long.190/
>
> [2] https://aclanthology.org/2022.emnlp-main.296/
>
> [3] https://arxiv.org/abs/2309.14556
>
> [4] https://www.ijcai.org/proceedings/2024/0003.pdf
>
> [5] https://arxiv.org/abs/2307.05300
>
> - *Examples of model outputs:* We are happy to include examples of model outputs. We will re-work Figure 2 to include the beginning of some stories and add complete examples in the appendix.
>
> - *About automated writing systems:* We will add more reflection on the use of automated systems for complex writing tasks. One of the reasons for developing the Agents’ Room framework was to move away from one-step black-box generation, in order to facilitate integrating a human-in-the-loop via a shared editable scratchpad. Previous work in the context of summarization ([1], [2])
>             has shown that while users have different modes of interacting with LLMs (desiring more    or less input from the AI), they overall expressed wanting more control over the generation process. A multi-step writing workflow with intermediate stages expressed in natural language that users can read and understand is a step in that direction. However, we leave designing an interactive user interface to assess collaborative writing in the context of our storytelling application to future work [3].
>
> [1] https://arxiv.org/abs/2206.06383
>
> [2] https://arxiv.org/abs/2206.14863
>
> [3] https://dl.acm.org/doi/pdf/10.1145/3635636.3656201
>
> **Answers to questions**
>
> 1. The human writers all have some type of advanced background in writing. Many have graduate degrees in English, some have professional writing experience, and many are former teachers.
> 2. Our human evaluators are better at spotting undesirable repetitions, inconsistencies, or incoherences than the LLM evaluator, so some discrepancies between both are expected. Our human participants found AR_write and AR_plan+write to be very similar when asked to rate the stories in the Overall dimension, whereas AR_write is somewhat better in terms of plot, creativity, and development, but worse in terms of language (see Figure 2). Although we did not elicit feedback on individual dimensions, we did ask participants to comment on the quality of the stories produced by our system, and possibly on aspects our rating system did not cover. We paste some of this feedback below and we will also include it in the appendix of our paper:
>
> ```
> "The task was interesting, but over time, I found the language redundant. There seemed to be a go-to vocabulary list utilized in the majority of the stories, phrases used time and again, making the output somewhat predictable."
>
> "It was interesting to see what kind of fictional narrative the model would generate. Most of the stories seemed to be written at a seventh grade level. The stories didn't stray too far from the input and for the most part were grammatically correct. There were at times, instances of repetitiveness, including entire paragraphs, that made me wonder what the model was doing."
>
> "The stories showed some promise, but often fell into the same pitfalls of loops or sudden tone discordance…"
> ```

---

> > ### Author Response · Authors · 2024-11-20
> > **We provide additional experiments with an open model**
> >
> > Following the reviewer’s suggestion, we reproduced our experimental setup with an open model, Gemma 2 9B.
> >
> > We ran two additional experiments:
> > - E2E: An end-to-end zero-shot baseline with Gemma 2 9B as model backbone.
> > - Agents’ Room (ours): A zero-shot multi-agent setup (with both planning and writing agents) with Gemma 2 9B as model backbone.
> >
> > While we consider human-based evaluation our primary means of evaluation, the LLM evaluator helps us assess overall system-level trends. In the following, we report pairwise win rate (proportion of examples on which our Agents’ Room framework performed better than the E2E baseline) according to our LLM evaluator:
> >
> > |  | overall  |  plot |  creativity |  development  | language use |
> > |---|---|---|---|---|---|
> > |  Agents’ Room (Gemma 2) vs E2E (Gemma 2) | 80.00  |  67.27 | 84.62  | 83.33  | 77.78 |
> >
> > Even with the Gemma 2 model backbone, we observe that our Agents’ Room (AR) framework greatly outperforms the end-to-end baseline across all dimensions of evaluation.
> >
> > We believe that with these additional experiments we have now addressed all the reviewer’s questions and comments.

---

> > > ### Author Response · Authors · 2024-11-22
> > >
> > > Dear reviewer, as the discussion period is nearing its end, we would like to inquire if you have had the opportunity to review our responses to your comments. We value your feedback very much and we hope the additional experiments we provided addressed your comments.

---

> > > > ### Comment · Reviewer_8uSS · 2024-11-26
> > > >
> > > > Thank you for running these additional experiments (as well as the additional experiments for Reviewer pcJk). I do strongly believe that this paper is above the acceptance threshold (as my previous score stated). To raise my score to "8: accept, good paper", I would need to see how these changes look in the context of the actual paper. Still, I would like the AC to know that I think the authors did a good job of addressing weaknesses raised by Reviewer pcJk (who has given the lowest score).

---

### Meta-Review · Area_Chair_h1ar · 2024-12-19

**Metareview:**

The paper presents a method to address the issue of long form narrative generation by breaking the task down into those that can be performed by specialized agents. The reviewers generally agree that the technical depth, motivation, and presentation are of sufficient quality to warrant publication. That said, reviewer 8uSS especially raises some great points in their weaknesses section eg advocating for the the overall framing of the paper to be focused on fiction writing as opposed to a generalized framework that I agree with and hope to see in a camera ready.

**Additional Comments On Reviewer Discussion:**

Only reviewer pcJk recommended a rejection and they also raised their scores after rebuttal though it seems to me that they could have raised it higher given all of their questions have been answered. The other two reviewers are clear that the paper is above the acceptance threshold.

---

### Decision · Program_Chairs · 2025-01-22

Accept (Poster)